# ReVeal: Self-Evolving Code Agents via Reliable Self-Verification

**Yiyang Jin**[2]* **Kunzhao Xu**[3]* **Hang Li**[1] **Xueting Han**[1]†
**Yanmin Zhou**[2] **Cheng Li**[3] **Jing Bai**[1]

[1]Microsoft Research Asia  [2]Tongji University  [3]University of Science and Technology of China

## ABSTRACT

Reinforcement learning with verifiable rewards (RLVR) has advanced the reasoning capabilities of large language models. However, existing methods rely solely on outcome rewards, without explicitly optimizing verification or leveraging reliable signals from realistic environments, leading to unreliable self-verification and limited test-time scaling. To address this, we widen the verification–generation asymmetry by explicitly optimizing self-verification, making it a reliable driver of deeper test-time scaling. We introduce **ReVeal**, a multi-turn **Re**inforcement learning framework that evolves code generation through self-**Ve**rification and tool-based ev**al**uation. ReVeal structures long-horizon reasoning as iterative generation–verification turns and incorporates TAPO for turn-level credit assignment, fostering the co-evolution of code and test generation. At inference, this strengthened self-verification enables the model to use self-constructed tests and tool feedback to continuously evolve code for **20+** turns on LiveCodeBench despite training on only three. It also significantly improves *Pass@k*, indicating stronger exploration that expands the reasoning boundaries of the base model. These findings highlight the promise of ReVeal as a scalable paradigm for RL training and test-time scaling, paving the way for more robust and autonomous AI agents. Code is available at https://ReVeal.github.io/.

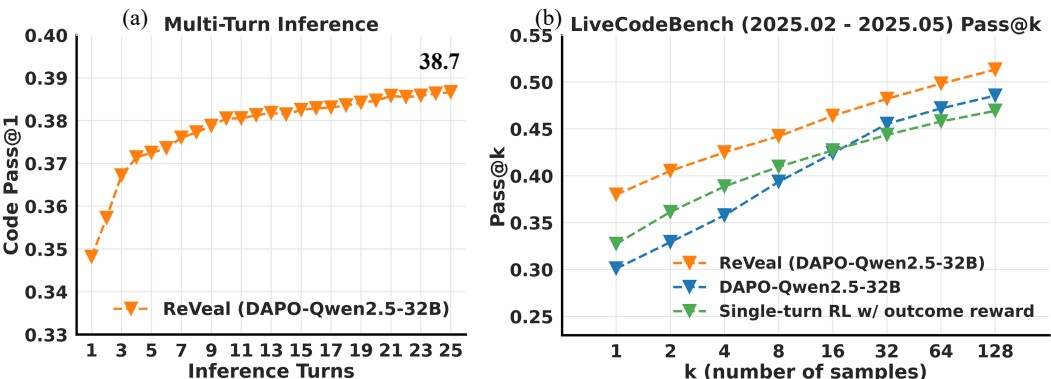

Figure 1: Performance of ReVeal on LiveCodeBench V6. (a) ReVeal enables effective test-time scaling, with Pass@1 accuracy improving from 34.8% at turn 1 to 38.7% at turn 25. (b) ReVeal (max_turn=10) consistently outperforms both the base model and the RL baseline in Pass@k, expanding the base model's reasoning boundaries, which the RL baseline fails to achieve.

## 1 INTRODUCTION

Reinforcement learning with verifiable rewards (RLVR) has recently shown strong potential to enhance the reasoning abilities of large language models (LLMs) (DeepSeek-AI et al., 2025; OpenAI).

*Equal contribution. Work was done during the internship at MSRA.

†Project leader; Correspondence to: Xueting Han <chrihan@microsoft.com>.

A key factor behind this success is the emergence of reflection and self-verification, which allow models to iteratively refine their reasoning. Recent analyses identify the *verification-generation asymmetry* (i.e., easier to verify than to solve) as the underlying mechanism for these improvements and a key driver of test-time scaling (Wei, 2025; Setlur et al., 2025). However, current RLVR methods rely solely on outcome rewards without explicitly optimizing verification. This leads to unreliable self-verification, where models often produce verbose, uninformative reflections or random guessing on hard problems, and limits the effectiveness of test-time scaling: prior studies show that reasoning performance plateaus once test-time compute exceeds the training horizon (Setlur et al., 2025).

Complex problem-solving, such as competitive programming, typically requires multiple iterations of verification and revision rather than being solved in a single attempt, making accurate feedback essential to guide refinement. This highlights the need for verification-driven multi-turn reasoning. Prior work has attempted this either by training a separate critic model to assess each attempt—without leveraging tool feedback and at the cost of added inference-time complexity (Xie et al., 2025)—or by relying on execution feedback against pre-existing public tests, which are rarely available in real-world scenarios (Gehring et al., 2025). As a result, these methods provide limited and non-generalizable verification, leaving self-verification unreliable and limiting sustained improvement.

To address these limitations, we propose **Re-Veal**, a multi-turn RL framework that *explicitly optimizes self-verification*, thereby widening the verification-generation (V-G) asymmetry and fostering co-evolution of both capabilities during training. This enables models at inference to obtain reliable verification signals from realistic environments and iteratively refine their solutions, without needing to rely on pre-existing tests. The widened V-G gap allows verification to drive sustained improvements in generation, ultimately enabling deeper test-time scaling (Figure 2).

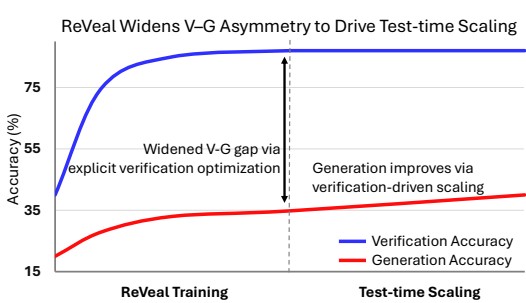

Figure 2: ReVeal expands the V-G gap.

Concretely, ReVeal structures long-horizon reasoning into iterative generation and verification turns. At each turn, the model generates candidate code and *self-verifies* its correctness by constructing test cases and invoking external tools (e.g., a Python interpreter) for execution. This closed loop yields actionable verification signals and fine-grained feedback, allowing the model to identify errors, revise strategies, and progressively refine its output across turns. For training, we attach *dense, turn-level rewards* that directly supervise both code quality and verification accuracy. To ensure robustness, ReVeal employs a *Turn-Aware Policy Optimization (TAPO)* tailored for the generation-verification interplay, assigning credit at the turn granularity and preventing reward gaming (e.g., generating trivial code to hack verification rewards). Unlike outcome-only RL methods, ReVeal makes verification itself an optimization target, turning verification signals into reliable drivers of improvement.

We evaluate ReVeal on the challenging LiveCodeBench benchmark (Jain et al., 2024). Notably, despite being trained on only three reasoning turns, ReVeal sustains continuous refinement for over 20+ inference turns, showing robust extrapolation beyond its training horizon and tackling problems previously unsolved. Furthermore, ReVeal significantly outperforms the base model in Pass@k by leveraging verification signals and tool feedback to guide more effective exploration, achieving an expansion of the underlying model's reasoning boundaries that standard RL methods fail to reach. These results validate ReVeal as not only a practical framework for self-evolving code agents, but also as a general RL paradigm for tasks with verification-generation asymmetry, where explicitly optimizing verification unlocks reliable long-horizon reasoning.

## 2 Methods

### 2.1 ReVeal Framework

#### 2.1.1 Iterative Generation–Verification Loop

ReVeal organizes long-horizon reasoning into an interleaved *generation-verification* loop with tool execution feedback, where verification itself is explicitly optimized to provide reliable signals for

multi-turn refinement. As illustrated in Figure 3, we use a single policy for both generation and verification to reduce system complexity and cost and to enable cross-capability transfer, so that solutions and their verification strategies co-evolve under a shared training scheme. In the code-generation setting, *generation* produces candidate code, whereas *verification* synthesizes and executes tests to assess correctness. Fine-grained feedback from tool execution (e.g., Python interpreter) is appended to the rollout and conditions the next turn. The loop continues until a valid solution is found or a turn budget $K$ is reached, enabling progressive refinement without external critics or predefined test cases.

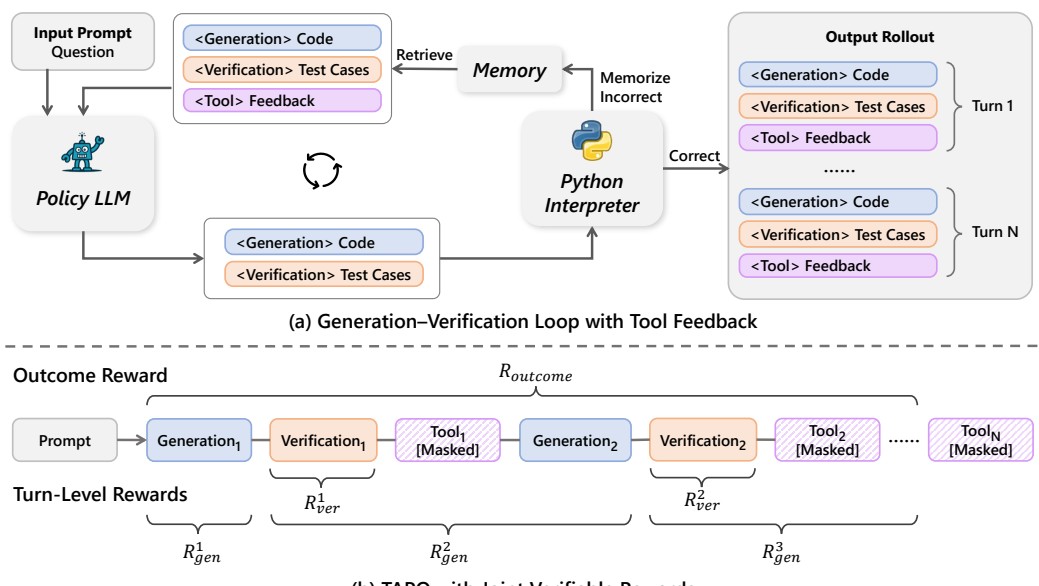

(a) Generation–Verification Loop with Tool Feedback

(b) TAPO with Joint Verifiable Rewards

Figure 3: Illustration of ReVeal. (a) Iterative generation-verification loop with tool feedback. (b) TAPO with joint verifiable rewards: outcome, generation, and verification rewards.

Table 4 illustrates a multi-turn rollout under ReVeal's structured prompting, which decouples generation, verification, and tool feedback into distinct segments. At each turn, the policy first reasons thoroughly and explores diverse reasoning patterns freely, then emits structured outputs: executable code in `<generation-answer>` and executable tests in `<verification-answer>`. As shown in the case study, after producing candidate code the model begins verification: it hypothesizes potential failure modes and edge conditions to propose diverse test cases. The `<tool-feedback>` section then records execution results, including runtime errors, invalid test cases, as well as the expected output, actual output, and pass/fail judgment for each valid test case. Based on this feedback, the model interprets traces and error messages, diagnoses underlying causes, and adjusts both its candidate code and its verification plan in the next turn. Full prompting and feedback templates are provided in Tables 5 and 6.

### 2.1.2 TOOL-AUGMENTED VERIFICATION

The interaction with external tools provides reliable, fine-grained supervisory signals that condition subsequent reasoning and enable systematic refinement of both code and verification strategies across turns. More importantly, tool interaction broadens exploration during reinforcement learning by revealing concrete failure modes, steering the policy into promising regions of the search space beyond a single attempt and helping it escape local optima. Empirically (see §3.3), this yields consistently higher Pass@k than the base model.

During RL training, the `<tool-feedback>` section is excluded from the loss and used only as contextual input, which stabilizes optimization while preserving coherent multi-turn rollouts. To ensure feedback quality during training, we adopt a filtering mechanism: model-generated test cases are executed on candidate code only if they are verified against a golden solution. This guarantees that execution traces provide legitimate supervision, thereby improving feedback precision and guiding exploration toward correct solutions. At test time, no golden reference is available; all generated

test cases are executed, making verification fully autonomous. This places a strong demand on the model's ability to generate high-quality tests. To meet this demand, ReVeal adopts a novel RL algorithm that incentivizes diverse and reliable test construction.

## 2.2 TURN-AWARE RL FOR THE GENERATION-VERIFICATION PARADIGM

Prior RLVR methods rely on outcome-only signals to optimize an entire long reasoning trace, but this provides imprecise credit to intermediate verification and often degenerates into blind reflection. Yet one may ask: *can the current paradigm fully sustain reliable verification and deeper test-time scaling?* Verification, however, is a non-trivial task: with well-designed verifiable rewards, a task can often be solved effectively. This motivates ReVeal to explicitly optimize verification with hard-to-hack rewards, which widen the verification-generation gap. At test time, this asymmetry becomes an asset: easier and more reliable verification signals can effectively guide the harder generation process to evolve over many turns.

### 2.2.1 JOINT VERIFIABLE REWARDS

To jointly train generation and verification, ReVeal decomposes the reward into three complementary components (Fig. 3b): an *outcome reward* supervising the final solution, a *generation reward* capturing improvements across generation turns, and a *verification reward* evaluating the quality of generated tests. This design naturally links the two roles in a co-evolutionary loop.

**Outcome reward.** The outcome reward shapes the entire reasoning chain by the final solution quality:

$$r_{\text{outcome}} = r_{\text{format}} + r_{\text{passrate}}, \tag{1}$$

where the format reward $r_{\text{format}}$ ensures that the model produces well-formed generation and verification blocks,

$$r_{\text{format}} = \begin{cases} 1, & \text{if the output format is correct,} \\ -1, & \text{otherwise,} \end{cases} \tag{2}$$

and $r_{\text{passrate}} = 5 \times passrate$ measures final code accuracy with $passrate \in [0, 1]$, giving $r_{\text{outcome}} \in [-1, 6]$. The format reward ensures that the output follows the prescribed generation-turn and verification-turn tags, so that we can reliably identify each turn and assign the correct turn-level reward, and that the code and test-case blocks satisfy the required format, allowing us to reliably extract code and tests for tool execution.

**Generation reward.** For each generation turn $k$ (odd), we compute the pass rate $r_{\text{passrate}}^k$ of the code produced and define:

$$r_{\text{gen}}^k = \begin{cases} r_{\text{passrate}}^1, & k = 1, \\ abs \cdot r_{\text{passrate}}^k + imp \cdot \left(r_{\text{passrate}}^k - r_{\text{passrate}}^{k-2}\right), & k \geq 3, \end{cases}$$

where $abs$ and $imp$ weight absolute accuracy and iterative improvement. We set $abs = 0$, $imp = 1$ so that the reward encourages real improvements in code accuracy across turns.

**Verification reward.** For each verification turn $k$ (even), we reward the proportion of generated tests that succeed when executed on a golden code:

$$r_{\text{ver}}^k = \frac{\#\{\text{test cases in turn } k \text{ that pass}\}}{\#\{\text{test cases generated in turn } k\}}. \tag{3}$$

### 2.2.2 TURN-AWARE POLICY OPTIMIZATION

**Preliminaries.** Our algorithm builds on the Proximal Policy Optimization (PPO) framework (Schulman et al., 2017), an on-policy actor-critic method that optimizes a clipped surrogate objective for stable updates. PPO typically estimates token-level advantages using Generalized Advantage Estimation (GAE) (Schulman et al., 2018):

$$\hat{A}_t^{\text{GAE}(\gamma,\lambda)} = \sum_{l=0}^{\infty} (\gamma\lambda)^l \left(r_{t+l} + \gamma V_{t+l+1} - V_{t+l}\right), \tag{4}$$

where $\gamma \in [0, 1]$ is the discount factor and $\lambda \in [0, 1]$ controls the bias-variance trade-off.

**Turn-Aware Policy Optimization.** Building on our structured reward design, we introduce *Turn-Aware Policy Optimization* (TAPO), which preserves the PPO actor-critic framework but only modifies the advantage estimator: instead of the standard GAE-based advantages, TAPO uses a turn-aware return to construct the advantage estimates. TAPO leverages the critic to efficiently bootstrap from both token-level Monte Carlo returns and turn-level returns, enabling stable learning across these two reward granularities.

1. **Token-level return.** We set $\lambda = 1$ and $\gamma = 1$ (pure Monte Carlo). For a response of length $T$, we define the token-level rewards as $r_T = r_{\text{outcome}}$ and $r_t = 0$ for all $t < T$. For token step $t$:

$$R_t = \sum_{l=0}^{T-t} r_{t+l} \; = \; r_t + R_{t+1}, \quad R_{T+1} = 0. \tag{5}$$

2. **Turn-level return.** To mitigate adversarial reward gaming (e.g., generating trivial code that hacks the verification reward), we introduce a *turn-level* return tailored to the generation-verification interplay. Specifically, (i) each generation reward is assigned both to its own generation turn and to the immediately preceding verification turn, and (ii) each verification reward is confined strictly to its own verification turn. This design prevents reward hacking by ensuring that generation turns are rewarded solely based on code quality, rather than verification success. Let $\{t_1, \ldots, t_K\}$ denote the token indices at which each turn ends (alternating generation and verification), and define:

$$R^{\text{turn}}(t_k) = \begin{cases} r_{\text{gen}}^k, & \text{if turn } k \text{ is generation,} \\ r_{\text{ver}}^k + R^{\text{turn}}(t_{k+1}), & \text{if turn } k \text{ is verification,} \end{cases} \quad R^{\text{turn}}(t_{K+1}) = 0. \tag{6}$$

For token $t$, let $\tau(t) = \min\{t_k \mid t_k \geq t\}$ and define

$$R_t^{\text{turn}} = \begin{cases} R^{\text{turn}}\big(\tau(t)\big), & \text{if } \tau(t) \text{ exists,} \\ 0, & \text{otherwise.} \end{cases} \tag{7}$$

3. **Turn-aware return.** The final return combines the two levels:

$$\widetilde{R}_t = R_t + R_t^{\text{turn}}, \qquad A_t = \widetilde{R}_t - V_t, \tag{8}$$

where $V_t$ is the critic model's estimate at step $t$. These advantages $A_t$ then replace the standard GAE estimates in the PPO objective, completing the TAPO update.

**Discussion.** TAPO provides sharper supervision than outcome-only methods by explicitly assigning credit at both token and turn levels. It integrates outcome rewards, which keep the process aligned with final correctness, and turn-level signals, which provide dense supervision for progressive refinement. This structure establishes a feedback loop: stronger tests expose errors that drive code improvements, which are then reinforced by the generation reward, while improved code raises the bar for verification, pushing the model to generate richer and more challenging tests. By design, TAPO prevents reward gaming and turns this loop into stable co-evolution of code and tests. Crucially, TAPO is a *general* credit-assignment algorithm, applicable to any reasoning task with verifiable rewards for both generation and verification.

## 3 EXPERIMENTS

### 3.1 SETTINGS

**Dataset** We construct our training dataset from TACO (Li et al., 2023), a large-scale corpus comprising 26,443 algorithmic programming problems sourced from competitive programming platforms such as LeetCode (LLC, 2015) and Codeforces (Codeforces, 2025). Each problem consists of a natural language description, golden solutions, and multiple test cases.

To address noise in the raw dataset, we first filter out problems containing unsupported content types, specifically those tagged with interactive or image elements. To ensure testability and correctness, we process two types of test case format, function-based tests and standard input/output tests, into a unified structure compatible with our code execution environment. We then execute each test case against the first available golden solution in our execution environment. Problems where the golden code fails to pass all associated test cases are discarded. After preprocessing, we retain a high-quality dataset of 11,151 problems for training and 509 problems for testing.

Table 1: Performance comparison of ReVeal with baseline methods on LiveCodeBench V6 and CodeContests. Pass@1 indicates the success rate; $\Delta_\uparrow$ and $\Delta_\downarrow$ represent the percentages of incorrect solutions corrected and correct solutions degraded after revision, respectively.

| Model | LiveCodeBench V6 | | | CodeContests | | |
|---|---|---|---|---|---|---|
| | Pass@1 | $\Delta_\uparrow$ | $\Delta_\downarrow$ | Pass@1 | $\Delta_\uparrow$ | $\Delta_\downarrow$ |
| *Existing Baselines* | | | | | | |
| Qwen2.5-32B-Instruct | 24.8 | - | - | 13.3 | - | - |
| DAPO-Qwen-32B | 31.1 | - | - | 18.5 | - | - |
| Qwen2.5-Coder-32B-Instruct | 29.5 | - | - | 14.6 | - | - |
| w/ critic×5 Qwen2.5-Coder | 29.6 | 2.14 | 3.04 | - | - | - |
| w/ critic×5 GPT-4o | 32.9 | 4.82 | 2.50 | - | - | - |
| w/ critic×5 CTRL | 33.4 | 3.75 | 0.89 | - | - | - |
| *RL based on DAPO-Qwen-32B* | | | | | | |
| Single-turn RL | 32.8 | - | - | 21.0 | - | - |
| ReVeal×25 | **38.7** | **7.50** | **0.0** | **33.6** | **15.69** | **0.0** |
| *Ablation Study: TAPO with Joint Verifiable Rewards* | | | | | | |
| ReVeal×8 w/ outcome reward | 36.1 | 4.69 | 1.32 | 27.4 | 9.24 | 2.36 |
| ReVeal×8 w/ TAPO with joint rewards | 37.7 | 5.62 | 0.0 | 30.4 | 12.30 | 0.0 |

**Models and Training Details** We adopt DAPO-Qwen-32B (Yu et al., 2025) as our base model, which is reinforced with mathematical data, and we continue RL training on code datasets to adapt its reasoning capabilities to coding tasks. Our models are trained using Verl (Sheng et al., 2024) framework on 8/16 AMD Mi300x GPUs. The RL training process follows the hyperparameter settings listed in Table 2.We set the maximum number of turns to 3 during RL training.

**Evaluation** We evaluate ReVeal on two code-generation benchmarks: LiveCodeBench (LCB) V6 (2025.02–2025.05) (Jain et al., 2024) and CodeContests (Li et al., 2022). The evaluation process follows the hyperparameter configuration specified in Table 3. Although training is performed with a maximum of 3 turns, we evaluate the model under extended turn settings (8 and 25 turns) to assess its generalization to longer reasoning horizons and test-time scaling performance.

We use Pass@1 to measure the success rate of the model's final code solutions. To evaluate the model's verification and self-correction capabilities, we introduce two additional metrics: $\Delta_\uparrow$ denotes the fraction of initially incorrect solutions that become correct after revision, and $\Delta_\downarrow$ denotes the fraction of initially correct solutions that become incorrect after revision. In line with recent work (Yue et al., 2025), we use Pass@k up to $k = 128$ to assess whether ReVeal can push the reasoning boundaries beyond the base model, with at most 10 generation–verification turns per example.

**Memory Mechanism for Context Management** To improve inference efficiency under extended multi-turn rollouts, we use a short-term memory mechanism that retains only the last three turns as context, which prevents excessive context growth without hurting accuracy (detail in Appendix C).

**Code Execution Tool** We use Code Judge[1] as our code execution environment. Code Judge supports both function-based and standard input-output test case formats through a consistent interface. Designed for scalability and robustness, it enables efficient long-batch execution through multiprocessing and provides reliable code evaluation.

**Baselines** We compare ReVeal against following baselines: (1) *Base*: base models without code-specific RL training; (2) *CTRL (Xie et al., 2025) + Qwen2.5-Coder-32B-Instruct*: five-turn critic–revision with a dedicated critic model; results cited from the original paper (evaluated on LCB 24.08–24.11); (3) *Single-turn RL with outcome reward*: RL with outcome-only rewards under standard <think>-<answer> prompting template without any external tool calls.

---

[1] https://github.com/0xWJ/code-judge

## 3.2 MAIN RESULTS

Table 1 shows that single-turn RL (outcome-only, no explicit optimization of self-verification or tool use) improves Pass@1 over the base models. ReVeal goes further by explicitly optimizing verification and enabling deeper inference, it surpasses the single-turn RL baseline by a wide margin. Beyond deeper-turn gains, ReVeal also achieves higher Pass@1 at turn 1 (34.8%) than the single-turn RL baseline under equal inference budget on LCB V6, indicating that multi-turn training (3 turns) transfers exploration benefits into a stronger policy and that increasing training depth may further amplify gains.

ReVeal significantly outperforms critic-based methods such as CTRL. While critic models tailored for code tasks can be paired with policy models for multi-turn critique and revision, ReVeal employs a single policy model that self-verifies and iteratively refines its own outputs, yet achieves superior results, highlighting the benefit of jointly optimizing generation and verification. Specifically, ReVeal attains larger correction rates with near-zero degradation, demonstrating highly robust and reliable capabilities in self-verification, critique, and revision. (CTRL numbers are cited from earlier LCB version; see Table 7 for a V5 comparison on Qwen2.5-32B-Instruct.)

Ablation studies confirm the benefit of TAPO with joint verifiable rewards: at the same turn budget it yields higher Pass@1, increases $\Delta_\uparrow$, and suppresses $\Delta_\downarrow$ compared to outcome-only training. In contrast, outcome-only rewards exhibit higher $\Delta_\downarrow$, indicating that insufficiently optimized verification can drive incorrect revisions.

**Extended Evaluations and Analyses.** To validate the effectiveness and scalability across models, we evaluate ReVeal on another base model, Qwen2.5-32B-Instruct. The detailed results are in Appendix D (see Table 7 and Figure 5). ReVeal outperforms the single-turn RL baseline by 4.1%, which demonstrates the effectiveness of ReVeal. To demonstrate that the performance improvements are statistically significant, we repeat the experiment 8 times and report the $mean \pm std$ in the Appendix Table 8. Across models of varying capability, ReVeal remains effective. With the stronger DAPO-Qwen-32B backbone, ReVeal unlocks greater headroom: accuracy continues to improve with deeper inference turns and surpasses outcome-only RL by a wider margin. This underscores ReVeal's potential on stronger backbones.

To further validate the generalizability of ReVeal and the specific contributions of its components, we provide extensive additional evaluations and analyses in Appendices E to K. Specifically, evaluations on Qwen3-4B-Instruct demonstrate ReVeal's effectiveness on smaller model scales and analyze the impact of varying training horizons (Appendix E, H). We also provide ablations to decouple the gains from our multi-turn framework versus RL training (Appendix F) and analyze the TAPO reward components (Appendix G). Beyond the main results, ReVeal's robust performance is confirmed on additional benchmarks like HumanEval+ and MBPP+ (Appendix I). Furthermore, a gradient-based incentive analysis for TAPO is provided in Appendix J. Finally, we discuss the current limitations and future directions in Appendix K.

## 3.3 ANALYSIS

**ReVeal Enables Test-time Scaling into Deeper Inference Regimes.** As shown in Figure 1 (a), ReVeal enables effective test-time scaling through iterative generation and verification. Although the model is trained with a maximum of three reasoning turns, it continues to improve its solutions when more turns are allowed at inference time, leading to progressively higher code accuracy. For instance, Pass@1 increases from 34.8% at turn 1 to 36.7% at turn 3, and further rises to 38.7% by turn 25 for LiveCodeBench. This compellingly demonstrates how reliable self-verification and iterative environment feedback can enable compute scaling into deeper inference regimes, allowing ReVeal to solve previously intractable problems and evolve novel solutions. As a result, ReVeal supports self-improvement beyond the training horizon, enabling strong generalization in long-horizon reasoning during inference. Furthermore, these newly discovered solutions can be distilled back into the code LLM to further enhance its reasoning capabilities through continued training.

**ReVeal Pushes Beyond the Reasoning Boundaries of the Base Model.** We compare DAPO-Qwen-32B and single-turn RL baseline with ReVeal using Pass@k metrics on LiveCodeBench. As shown in Figure 1 (b), the RL baseline outperforms the base model when k < 32, but its performance

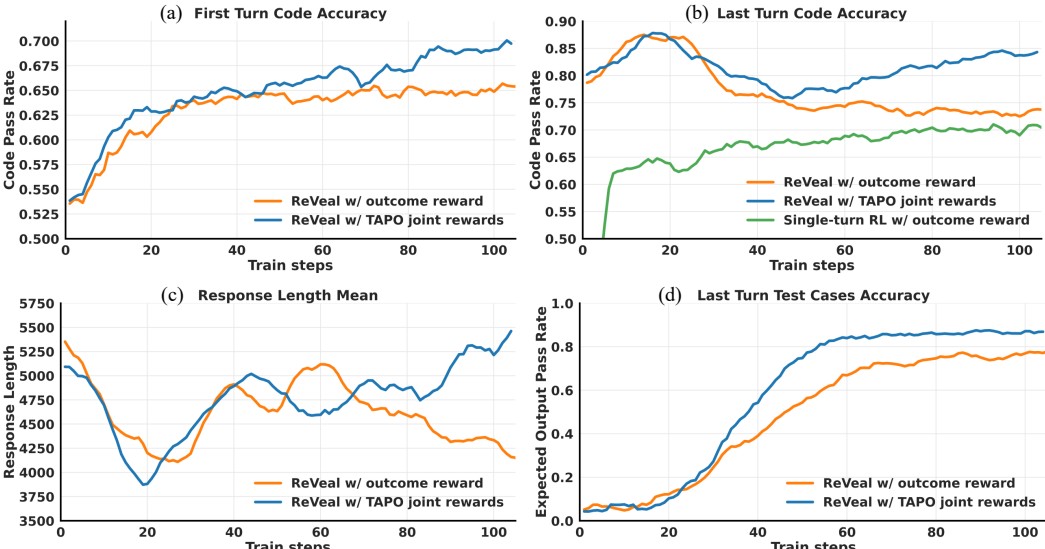

Figure 4: Training curves for (a) first-turn code accuracy, (b) last-turn code accuracy, (c) mean response length, and (d) last-turn test-case accuracy across three training methods. **Note on (b):** the dip before step 40 is due to expanded evaluation coverage: as format score reaches 0.9 around step 40, more problems enter the evaluation set, temporarily lowering accuracy.

gain gradually diminishes as k increases. In contrast, ReVeal consistently outperforms both the base model and the RL baseline across all k values from 1 to 128, demonstrating its ability to surpass the reasoning boundaries beyond the base model. We attribute this improvement to ReVeal's verification-driven exploration: tool-assisted verification provides targeted, execution-based feedback and precise judgments that guide the model to explore better solutions more effectively. With this enhanced exploration capability, the model continually self-evolves and grows beyond its initial reasoning capability during RL training. We believe this approach offers a promising path towards developing self-evolving agents with stronger reasoning capabilities.

**ReVeal Co-evolves the Model's Generation and Verification Capabilities.** Figure 4(b, d) illustrates the co-evolution of the model's code and test case generation capabilities. As shown in Figure 4(b), once the format is learned, final code accuracy steadily improves during training and significantly surpasses the single-turn RL baseline. Moreover, comparing Figure 4(a) and (b) reveals that final solutions consistently outperform those generated at Turn 1, with the performance gap widening over time. This trend indicates that as the model's verification ability strengthens, multi-turn refinement enables the exploration of better solutions, progressively enhancing its capacity to generate and refine code. After the format is learned, test-case accuracy rises substantially from about 50% at step 40 to nearly 88%, as shown in Figure 4 (d). Additionally, for correctly generated test cases, the model achieves over 85% accuracy in judging code correctness. This demonstrates that during inference, the model can reliably generate valid test cases and effectively leverage the tool to produce accurate verification signals, which are critical for continuous improvements in code quality. These results provide strong evidence that ReVeal jointly and effectively optimizes both generation and verification, enabling the model to evolve its reasoning capabilities throughout training.

**The Effectiveness of TAPO with Joint Verifiable Rewards.** As shown in Table 1, TAPO with joint rewards further enhances multi-turn performance compared to relying solely on outcome rewards. The training curves in Figure 4(a,b) show TAPO with joint rewards achieves more stable and consistent per-turn code gains, and Figure 4(d) shows it achieves higher test-case accuracy, indicating that explicitly optimizing verification yields higher-quality tests and more effective reasoning in code generation task. These benefits amplify in longer-sequence and harder verification scenarios. On the stronger DAPO-Qwen-32B backbone with longer chains, dense turn-level supervision yields larger gains than on Qwen2.5-32B-Instruct with much shorter chains (see Table 7 and Figure 5). This is because outcome-only signals are too coarse for extremely long chains, providing imprecise credit to intermediate verification steps. Furthermore, in more challenging verification scenarios, such

fine-grained supervision becomes increasingly essential, offering richer learning signals to enhance the model's verification capabilities.

## 4 RELATED WORK

### 4.1 TOOL-AUGMENTED REASONING

Recent advances in post-training for LLMs have led to substantial progress in improving reasoning capabilities (Chen et al., 2025b;a). Among these efforts, tool-augmented approaches enables large language models (LLMs) to leverage external tools, such as search engines or code interpreters, to overcome inherent limitations in domain knowledge and mathematical operations. Early approaches demonstrated the benefits of tool integration via prompt engineering (Yao et al., 2023; Chen et al., 2023; Shinn et al., 2023) and supervised fine-tuning (Gou et al., 2024). ReAct (Yao et al., 2023) and Reflexion (Shinn et al., 2023) interleave reasoning with acting or verbal self-critique to iteratively refine solutions under tool access. These approaches highlight the value of interactive signals, but they typically rely on prompt heuristics. More recently, multi-turn RL has been adopted to further enhance this capability on various reasoning tasks (Jin et al., 2025; Feng et al., 2025; Li et al., 2025). For example, Search-R1 (Jin et al., 2025) incorporates multi-turn interactions with a search engine to retrieve relevant contextual information during RL training. ReTool (Feng et al., 2025) and ToRL (Li et al., 2025) enable multi-turn code execution to support mathematical reasoning. Building on the promising potential of tool-integrated RL, Agent-R1 (Ouyang et al., 2025) introduces an open-source RL framework capable of supporting multi-turn, customization tool invocations.

Despite their effectiveness, most tool-augmented RL methods are predominantly outcome-driven: they rely on task success or failure as the sole training signal and do not explicitly optimize verification or assign credit across turns. Likewise, prompt-only agents lack turn-level, verifiable supervision, which can make self-verification unreliable on harder problems and limit sustained test-time improvement. Unlike prior tool-augmented works, ReVeal treats verification itself as a first-class optimization target alongside generation, and introduces Turn-Aware Policy Optimization rewards (TAPO) to provide fine-grained credit to both generation and verification turns. Our approach and prior tool-augmented methods are orthogonal and complementary, and can be naturally combined to further enhance reasoning capability.

### 4.2 SELF-VERIFICATION OF LLMS

Enabling LLMs to iteratively refine their outputs is critical for enhancing their reasoning capabilities. However, LLMs typically lack reliable self-judgment (Huang et al., 2024). One common solution is to introduce a separate critic model to verify the output of the policy model (Zhang et al., 2025; Xie et al., 2025). For example, CTRL (Xie et al., 2025) uses RL to train a critic model for code completion tasks. Although effective, these approaches incur the cost and complexity of maintaining and coordinating two distinct models.

An alternative strategy is to enable one single model to generate outputs and self-verify them. In mathematical reasoning, (Xiong et al., 2025) synthesizes long chains of thought that incorporate "self-reward" and "self-correction" signals as seed data for supervised fine-tuning, and then further enhances this ability via RL. In the code domain, execution feedback effectively verifies code correctness and provides useful information for fixing errors. RLEF (Gehring et al., 2025) performs multi-turn code generation and verification with an integrated code execution tool; however, it depends on publicly available test cases, limiting its applicability.

In contrast, ReVeal advances self-verification by having the model generate its own high-quality test cases on the fly. By explicitly crafting and executing these tests, ReVeal eliminates the dependency on pre-existing test suites and improves applicability to real-world software systems.

## 5 CONCLUSION

We presented REVEAL, a multi-turn reinforcement learning (RL) framework that makes verification a first-class optimization target alongside generation and organizes long reasoning chains into iterative generation–verification turns with tool feedback. Using TAPO with joint verifiable rewards, REVEAL

equips LLMs with strong verification capabilities and demonstrates the surprising power of enabling code LLMs to self-evolve—both during RL training, where it pushes boundaries beyond the base model, and at test time, where multi-turn generation and verification continually refine outputs, even up to 20+ inference turns. This compellingly demonstrates that REVEAL can enable compute scaling into deeper inference regimes, allowing it to solve previously intractable problems and evolve novel solutions. Furthermore, these newly discovered solutions can be distilled back into the code LLM to further enhance its reasoning capabilities through continued training.

Although we demonstrate REVEAL on code tasks, its general concept of generation–verification, TAPO, and turn-level reward design can be applied to any domain with verifiable rewards for both generation and verification and that exhibits verification asymmetry, offering a promising blueprint for future advances in self-improving, more robust, and autonomous AI agents.

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

# A  THE USE OF LARGE LANGUAGE MODELS

In preparing this manuscript, we used a large language model (LLM) solely for polishing the writing style and improving the clarity of the manuscript. The LLM was not used for generating research ideas, designing experiments, conducting analyses, or deriving results. All scientific contributions, including the conceptualization, methodology, experiments, and conclusions, were developed entirely by the authors.

# B  IMPLEMENTATION DETAILS

## B.1  HYPERPARAMETERS

Table 2 and Table 3 show the detailed hyperparameters we use during training and evaluation.

Table 2: RL Training Hyperparameters for ReVeal-Qwen2.5-32B-Instruct and ReVeal-DAPO-Qwen-32B

| Parameter | Qwen2.5-32B-Instruct | DAPO-Qwen-32B |
|---|---|---|
| Max Turn | 3 | 3 |
| Training Batch Size | 128 | 1024 |
| Mini-Batch Size | 16 | 256 |
| Learning Rate (Actor) | $5 \times 10^{-7}$ | $1 \times 10^{-6}$ |
| Learning Rate (Critic) | $1 \times 10^{-5}$ | $1 \times 10^{-5}$ |
| KL Coefficient | 0.0 | 0.0 |
| Maximum Prompt Length | 4,096 | 4,096 |
| Maximum Response Length | 8,192 | 12,288 |
| Maximum Tool Response Length | 4,096 | 4,096 |
| Temperature | 1.0 | 1.0 |
| Training Epochs | 4 | 10 |

Table 3: RL Evaluation Hyperparameters for Qwen2.5-32B-Instruct and DAPO-Qwen-32B

| Parameter | Qwen2.5-32B-Instruct | DAPO-Qwen-32B |
|---|---|---|
| Maximum Prompt Length | 4,096 | 4,096 |
| Maximum Response Length | 20,480 | 32,768 |
| Maximum Tool Response Length | 4,096 | 4,096 |
| Do Sample | True | True |
| Temperature for Pass@1 | 0.6 | 0.6 |
| Top_p for Pass@1 | 0.9 | 0.9 |
| Rollout for Pass@1 | 10 | 10 |
| Temperature for Pass@k | 0.6 | 0.6 |
| Top_p for Pass@k | 0.9 | 0.9 |
| Rollout for Pass@k | 128 | 128 |

## B.2  CASE STUDY OF REVEAL

Table 4 shows a detailed case of ReVeal.

## B.3  PROMPT TEMPLATES

Table 5 shows the comparison between the commonly used Think-Answer prompt and our Generation-Verification Prompt. Our prompt guides the model to continuously alternate between generation and verification until the correct answer is obtained. Additionally, to enable the extraction of code generated by the model for providing accurate training rewards, we instruct the model to enclose the code within python blocks.

Table 4: Case Study of ReVeal Performing Iterative Generation and Verification

**Question:**
The Two-dimensional kingdom has a regular army of $n$ people... The Two-dimensional kingdom has $m$ vests at its disposal, the $j$-th vest's size equals $b^j$. Help mobilize the Two-dimensional kingdom's army: equip with vests as many soldiers as possible.

**Response**:
**\<generation-think\>**
The problem can be solved using a dual circulation.
Here's a step-by-step approach to the solution:
...
Wait, I can use a greedy approach to solve this problem. Let me try the greedy algorithm by following steps:
...
**\</generation-think\>**
**\<generation-answer\>**
```python
def equip_soldiers(n, m, x, y, a, b):
        ...
```
**\</generation-answer\>**
**\<verification-think\>**
Now, we will verify the code with the provided test cases.
   - Example Input 1 should result in matching 2 pairs: (Soldier at index 1 with Vest at index 1) and (Soldier at index 3 with Vest at index 2). There is no possible matching for the remaining soldiers with the available vests within the given range.
Wait, I could add an extra test case for verification: more vests than soldiers...
**\</verification-think\>**
**\<verification-answer\>**
- Input:
```1 3 5```
- Expected Output:
```3 2```
...
**\</verification-answer\>**
**\<tool-feedback\>**
...
- Actual Output:
```3 1```
- Judgement:
Failed
...
**\</tool-feedback\>**

**\<generation-think\>**
The initial code provided did not successfully handle the actual output as expected. There is a discrepancy where the vest index tracked in the loop is not correctly being reset or updated...
Alternatively, I should create a list that keeps track of which vests have already been used and adjust the code logic to ensure each vest is used only once. Here is a revised version of the code:
...
**\</verification-answer\>**
**\<tool-feedback\>**
...
- Actual Output:
```3 2```
- Judgement:
Passed
...
**\</tool-feedback\>**

Table 5: Comparison Between Think-Answer Prompt and ReVeal Prompt

| Think-Answer Prompt | ReVeal Prompt |
|---|---|
| **system**
You are Qwen, created by Alibaba Cloud. You are a helpful assistant.
**user**
{question}

First think about the reasoning process in the mind and then provides the user with the answer. The reasoning process and answer are enclosed within \<think\> \</think\> and \<answer\> \</answer\> tags, respectively, i.e., \<think\> reasoning process here \</think\> \<answer\> answer here \</answer\>. Enclose your code within delimiters as follows.
```python
YOUR CODE HERE
```
**assistant** | **system**
You are Qwen, created by Alibaba Cloud. You are a helpful assistant.
**user**
{question}

First think through the reasoning process and write Python code to solve the problem, enclose your reasoning process in \<generation-think\> \</generation-think\> and present the code in
```python
Your code
```
within \<generation-answer\> \</generation-answer\> tags. After that, verify your code by generating test cases:

1. Extract sample test cases if the problem description includes them. When necessary, generate a small number of additional test cases to validate the correctness of the generated code.
2. Enclose your reasoning process in \<verification-think\> \</verification-think\> tags and enclose the final test cases and your verification conclusion within \<verification-answer\> \</verification-answer\> tags and wrap each test case using the following format:
- Input:
```
testcase input
```
- Expected Output:
```
expected testcase output
```
3. Note that for "Use Call-Based format" questions, the testcase input should use a function call format, e.g., fn_name(12, 12, 12).
**assistant** |

### B.4 TEMPLATES USED FOR TOOL FEEDBACK

Table 6 shows the mapping between execution results and hint templates: (1) for test cases that are verified as successful, we give a [Passed] signal in the judgement area; (2) for test cases that are verified as failed, we give a [Failed] signal in the judgement area; (3) for test cases that are verified as wrong, we give a clear feedback of [Wrong test case] for individual failures, or [No correct test cases generated] if all test cases are invalid; (4) for format error, we will give the feedback of formatting instructions to guide correct generation.

## C THE EFFECTIVENESS OF SHORT-TERM MEMORY

To prevent the interaction context from growing unbounded and to keep inference efficient, we use a short-term memory mechanism. The system keeps a rolling window of the last three generation–verification turns and discards older interactions. For those retained turns, it preserves the full content of code, test cases, and tool feedback. This design allows the model to reuse information from recent history and build on previous attempts, which helps reduce redundant exploration and speed up convergence.

We compare models with and without memory integration. The baseline model without memory provides complete historical information from all previous turns directly to the model, maintaining full contextual details throughout the interaction sequence. We evaluated ReVeal's impact on LiveCodeBench Pass@1 performance with and without memory. Test results indicate that introducing short-term memory does not cause a decline in Pass@1 (w/o memory 38.2% vs. w/ memory 38.3% at turn 15), and may even yield a slight performance boost. This sustained improvement capability highlights the memory mechanism's effectiveness in enabling continuous learning and adaptation within computational constraints.

Table 6: Tool Feedback Templates for Different Execution Result Types.

| Execution Results | Feedback |
|---|---|
| Success Test cases | - Input:
{input}

- Expected Output:
{expected output}

- Actual Output:
{actual output}

- Judgement
Passed |
| Failed Test cases | - Input:
{input}

- Expected Output:
{expected output}

- Actual Output:
{actual output}

- Judgement
Failed

- Failed Reason
{failed reason} |
| Wrong Test Cases | - Input:
{input}

- Expected Output:
{expected output}

- Actual Output:
{actual output}

- Judgement
Wrong test case.

No correct test cases are generated. |
| Error Format | No valid code because of the incorrect format. Write Python code again, and present the code in
```python
Your code
```
within <generation-answer> </generation-answer> tags. After that, verify your code by generating test cases:
1. Extract sample test cases if the problem description includes them...
2. Wrap each test case using the following format:
- Input:
```
testcase input
```

- Expected Output:
```
expected testcase output
``` |

# D    MORE EXPERIMENTS ON QWEN2.5-32B-INSTRUCT

**Main Results.**    To further validate the effectiveness and scalability of ReVeal across different base models, we additionally conduct experiments on Qwen2.5-32B-Instruct. As shown in Table 7, ReVeal achieves a 4.1% improvement in Pass@1 over the single-turn RL baseline. Moreover, using TAPO with joint verifiable rewards yields higher Pass@1 compared to ReVeal with outcome-only reward, while maintaining a higher $\Delta \uparrow$ and near-zero $\Delta \downarrow$. These results demonstrate that ReVeal remains effective and stable across backbones of varying capability.

**Training Curves.**    Figure 5 shows concurrent improvements in both code accuracy and test-case accuracy when using TAPO with joint rewards, indicating that explicitly optimizing self-verification leads to more reliable verification and drives stronger multi-turn refinement, echoing our main findings on DAPO-Qwen-32B.

**Significance.**    To demonstrate that the performance improvements are statistically significant, we repeated the experiment 8 times and reported the $mean \pm std$ in Table 8, confirming the significance of the gains.

Table 7: Performance comparison of ReVeal (Qwen2.5-32B-Instruct) with baseline methods on LiveCodeBench. Pass@1 indicates the success rate; $\Delta_{\uparrow}$ and $\Delta_{\downarrow}$ represent the percentages of incorrect solutions corrected and correct solutions degraded after revision, respectively.

| Model | LiveCodeBench V5 | | |
|---|---|---|---|
| | **Pass@1** | $\Delta_{\uparrow}$ | $\Delta_{\downarrow}$ |
| *Existing Baselines* | | | |
| Qwen2.5-32B-Instruct | 26.6 | - | - |
| DAPO-Qwen-32B | 29.6 | - | - |
| Qwen2.5-Coder-32B-Instruct | 30.5 | - | - |
|    w/ critic×5 Qwen2.5-Coder | 29.6 | 2.14 | 3.04 |
|    w/ critic×5 GPT-4o | 32.9 | 4.82 | 2.50 |
|    w/ critic×5 CTRL | 33.4 | 3.75 | 0.89 |
| *RL based on Qwen2.5-32B-Instruct* | | | |
| Single-turn RL | 33.9 | - | - |
| ReVeal×6 | **38.0** | **3.41** | **0.0** |
| *Ablation Study: TAPO with Joint Verifiable Rewards* | | | |
| ReVeal×6 w/ outcome reward | 37.1 | 2.98 | 0.0 |
| ReVeal×6 w/ TAPO with joint rewards | 38.0 | 3.41 | 0.0 |

Table 8: Significance test of ReVeal (Qwen2.5-32B-Instruct) on LiveCodeBench V5. $mean \pm std$ indicates the average code Pass@1 from 8 repeated experiments.

| Model | LiveCodeBench V5 |
|---|---|
| | $mean \pm std$ |
| ReVeal×6 turn w/ outcome reward | $37.09 \pm 0.31$ |
| ReVeal×6 turn w/ TAPO with joint rewards | $38.02 \pm 0.33$ |

# E    EXPERIMENTS ON QWEN3-4B-INSTRUCT

To cover models of different scales, we additionally evaluated ReVeal on smaller model Qwen3-4B-instruct (Yang et al., 2025). The results on Table 9 show that ReVeal brings substantial performance gains, far exceeding other baseline methods and enables deeper test-time scaling. This provides further evidence that ReVeal is not tied to a single scale (32B) and can generalize across different models.

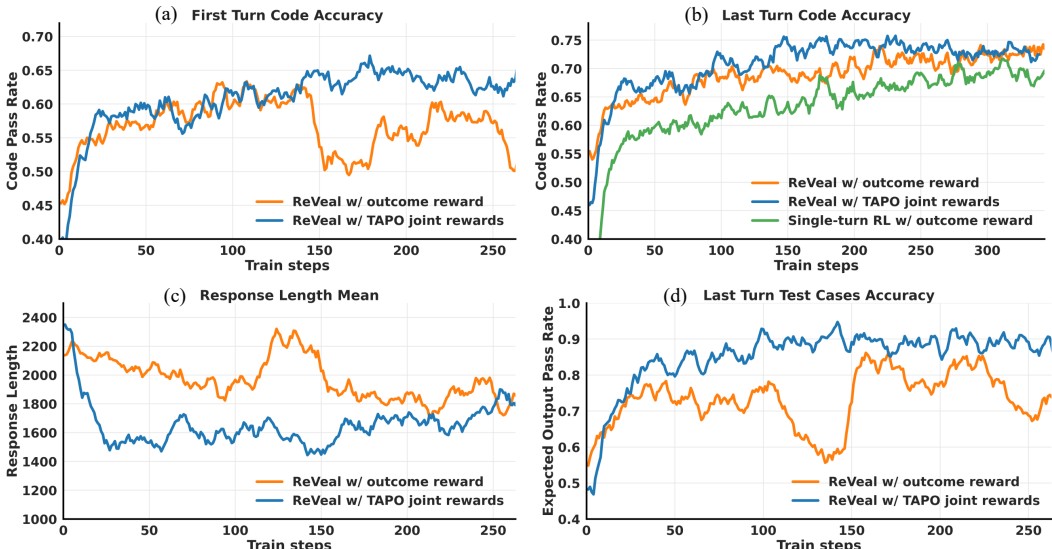

Figure 5: Comparison of code accuracy, test case accuracy, and response length across training for ReVeal (Qwen2.5-32B-Instruct) with turn-level rewards, ReVeal with outcome-only rewards, and single-turn RL without tool integration.

Table 9: Performance comparison of ReVeal trained based on Qwen3-4B-Instruct with different training turns. Pass@1 indicates the success rate.

| Model | LiveCodeBench V6 | CodeContests |
|---|---|---|
| | Pass@1 | Pass@1 |
| Qwen3-4B-Instruct | 33.1 | 24.3 |
| Single-turn RL | 39.0 | 26.9 |
| ReVeal (Max train turn=1) | | |
| ×1 turn | 37.3 | 26.6 |
| ×5 turn | 41.5 | 30.6 |
| ×8 turn | 41.7 | 30.7 |
| ReVeal (Max train turn=3) | | |
| ×1 turn | 40.6 | 28.5 |
| ×5 turn | 44.1 | 33.6 |
| ×8 turn | **44.5** | **33.9** |
| ReVeal (Max train turn=5) | | |
| ×1 turn | 38.6 | 28.4 |
| ×5 turn | 43.1 | 33.2 |
| ×8 turn | 44.0 | 33.5 |

## F    ABLATIONS: SEPARATING FRAMEWORK-LEVEL AND RL TRAINING GAINS

To disentangle the effect of the ReVeal multi-turn framework from the effect of ReVeal RL training, we evaluate several variants that progressively add components of ReVeal.

We first apply the ReVeal multi-turn framework to Qwen2.5-32B-Instruct without any RL training, using exactly the same ReVeal prompt format and code-execution tool. As shown in Table 10, this variant produces only modest gains and fails to sustain effective deep multi-turn refinement: performance improves slightly from 1 to 3 turns and then saturates or even drops. This indicates that simply changing the prompting scheme and enabling multi-turn tool feedback is not sufficient.

We then apply the same ReVeal multi-turn framework with code execution to stronger baselines such as DAPO-Qwen-32B and Single-turn RL at test time, again without additional RL under the ReVeal objective, these multi-turn variants either exhibit gains in the first few turns and then begin to degrade,

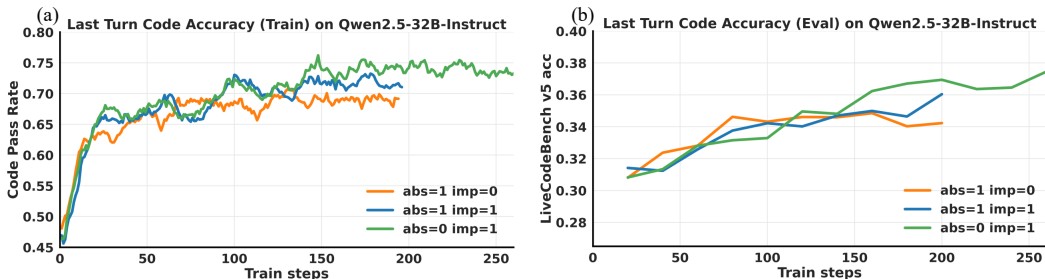

Figure 6: Effect of generation-reward coefficients $(abs, imp)$ on ReVeal (Qwen2.5-32B-Instruct). We plot (a) training code accuracy and (b) LiveCodeBench v5 accuracy across training for three settings of: $(1, 0)$, $(1, 1)$, and $(0, 1)$.

or show almost no improvement at all (Table 11). While this confirms that most of ReVeal's gains cannot be explained by multi-turn framework alone.

Finally, Table 11 reports two stronger ablations, ReVeal (no-gen RL) and ReVeal (no-ver RL), where we still use the ReVeal multi-turn framework but RL-train only one role: we pair a ReVeal-trained verifier with an untrained base generator (no-gen RL), or a ReVeal-trained generator with an untrained base verifier (no-ver RL). ReVeal (no-ver RL) yields gains in the first few turns but quickly saturates, whereas the full ReVeal model continues to improve with more turns and achieves higher $\Delta_\uparrow$ and lower $\Delta_\downarrow$. This indicates that ReVeal's explicit optimization of verification yields clear performance gains and enables deeper test-time scaling. ReVeal (no-gen RL) continues to improve with more turns even when paired with a relatively weak generator, suggesting that ReVeal's explicit optimization of verification is key to enabling deeper test-time scaling. Taken together, these ablations indicate that while the multi-turn framework and tool feedback are helpful, the main gains come from explicitly training by ReVeal RL objective.

Table 10: Performance of Qwen2.5-32B-Instruct under the ReVeal multi-turn framework, with and without RL training. Pass@1 indicates the success rate; $\Delta_\uparrow$ and $\Delta_\downarrow$ represent the percentages of incorrect solutions corrected and correct solutions degraded after revision, respectively.

| Model | LiveCodeBench V5 | | |
|---|---|---|---|
| | Pass@1 | $\Delta_\uparrow$ | $\Delta_\downarrow$ |
| ReVeal Multi-turn Framework w/o Training | | | |
| $\times 1$ turn | 26.3 | - | - |
| $\times 3$ turn | 27.5 | - | - |
| $\times 6$ turn | 27.4 | 2.40 | 1.31 |
| ReVeal Multi-turn Framework w/ Training | | | |
| $\times 1$ turn | 35.7 | - | - |
| $\times 3$ turn | 37.5 | - | - |
| $\times 6$ turn | 38.0 | 3.41 | 0.0 |

## G  TAPO REWARD ABLATIONS

**TAPO Reward Ablations**    Besides the reward-signal ablation comparing ReVeal (outcome-only) variant with the full joint-reward in Table 1, we additionally include two ReVeal variants on top of the outcome-only baseline: ReVeal (outcome+gen) and ReVeal (outcome+ver). As shown in Table 12, both variants outperform the outcome-only version, and the full ReVeal achieves the overall best results. This indicates that both $r_{gen}$ and $r_{ver}$ make useful contributions beyond the outcome reward: $r_{gen}$ supervises both generation and verification by encouraging exposing more informative failure modes and larger subsequent code improvements, while $r_{ver}$ supervises test case accuracy directly.

**Generation-Reward Ablation**    Our generation reward decomposes into an absolute term (abs, the code accuracy at the current turn) and an improvement term (imp, how much the current turn's

Table 11: Performance on LiveCodeBench V6 for configurations designed to disentangle the gains from the ReVeal multi-turn framework and from ReVeal RL training. We compare: (i) applying the multi-turn framework to existing baselines without ReVeal RL, (ii) ReVeal variants where only the generator or only the verifier is RL-trained (no-gen RL / no-ver RL), and (iii) full ReVeal with RL on both roles.

| Configuration | LiveCodeBench V6 | | |
|---|---|---|---|
| | **Pass@1** | $\Delta_\uparrow$ | $\Delta_\downarrow$ |
| *Existing Baselines + ReVeal Multi-turn Framework* | | | |
| Base Model + ReVeal multi-turn framework | | | |
| ×1 turn | 28.4 | - | - |
| ×5 turn | 32.8 | 4.53 | 0.75 |
| ×8 turn | 31.8 | 4.15 | 1.70 |
| Single-turn RL model + ReVeal multi-turn framework | | | |
| ×1 turn | 33.4 | - | - |
| ×5 turn | 32.8 | 1.53 | 1.36 |
| ×8 turn | 33.0 | 2.05 | 1.53 |
| *ReVeal with RL on one role* | | | |
| ReVeal (no-gen RL): Base model as Generator + ReVeal as Verifier | | | |
| ×1 turn | 29.0 | - | - |
| ×5 turn | 33.6 | 6.16 | 0.91 |
| ×8 turn | 33.8 | 6.39 | 0.91 |
| ReVeal (no-ver RL): ReVeal as Generator + Base model as Verifier | | | |
| ×1 turn | 34.6 | - | - |
| ×5 turn | 36.9 | 2.94 | 0.53 |
| ×8 turn | 36.8 | 4.25 | 1.26 |
| *ReVeal with full RL training* | | | |
| ReVeal | | | |
| ×1 turn | 34.8 | - | - |
| ×5 turn | 37.2 | 3.71 | 0.0 |
| ×8 turn | 37.7 | 5.62 | 0.0 |

code accuracy improves over the previous turn). In our default setting, we use abs = 0 and imp = 1; that is, we only use the change in code pass rate to reward the quality of the generated tests and the current-turn correction. In early experiments on Qwen2.5-32B-Instruct, we compared several weightings (abs/imp = 0/1, 1/1, 1/0) and observed 0/1 > 1/1 > 1/0 as shown in Figure 6, suggesting that emphasizing improvement over absolute accuracy leads to better multi-turn performance.

## H TRAINING-TURN COUNT ABLATION

We vary the training horizon to 1, 3, and 5 turns while keeping other settings fixed. As shown in Table 9, all ReVeal variants outperform the single-turn RL baseline; training with 3 turns yields much larger gain than 1-turn training, while increasing the horizon further to 5 turns brings no additional improvement, suggesting that the benefit saturates around 3-5 turns on our current (mostly solvable) training set. On this training set, mean rewards can reach 4.5, which means most problems are already solved within 3 generation-verification turns, so the additional 2 training turns provide limited extra signal on a small fraction of very difficult questions. We expect longer training horizons to be more beneficial on more challenging training data.

Importantly, for all training horizons (1, 3, and 5), the corresponding models continue to improve when we allow more inference turns than they were trained on, indicating that extrapolation beyond the training horizon is effective even when the model is trained with only a single turn.

Table 12: Performance comparison of ReVeal trained based on DAPO-Qwen-32B with different verifiable rewards. Pass@1 indicates the success rate.

| Model | LiveCodeBench V6 | CodeContests |
|---|---|---|
| | Pass@1 | Pass@1 |
| ReVeal (outcome only) | | |
| $\times 1$ turn | 32.7 | 20.0 |
| $\times 5$ turn | 35.9 | 26.1 |
| $\times 8$ turn | 36.1 | 27.4 |
| ReVeal (outcome+ver) | | |
| $\times 1$ turn | 32.3 | 17.3 |
| $\times 5$ turn | 36.6 | 26.5 |
| $\times 8$ turn | 36.8 | 27.8 |
| ReVeal (outcome+gen) | | |
| $\times 1$ turn | 30.7 | 17.2 |
| $\times 5$ turn | 37.1 | 26.6 |
| $\times 8$ turn | **38.0** | 28.2 |
| ReVeal Full (outcome+gen+ver) | | |
| $\times 1$ turn | 34.8 | 22.3 |
| $\times 5$ turn | 37.2 | 28.1 |
| $\times 8$ turn | 37.7 | **30.4** |

Table 13: Performance comparison of ReVeal (DAPO-Qwen-32B) with baseline methods on HumanEval+ and MBPP+. Pass@1 indicates the success rate.

| Model | HumanEval+ | MBPP+ |
|---|---|---|
| | Pass@1 | Pass@1 |
| *Existing Baselines* | | |
| Qwen2.5-32B-Instruct | 85.4 | 75.4 |
| DAPO-Qwen-32B | 86.0 | 73.8 |
| *RL based on DAPO-Qwen-32B* | | |
| Single-turn RL | 86.0 | 76.2 |
| ReVeal$\times 8$ | | |
| $\times 1$ turn | 85.2 | 77.9 |
| $\times 3$ turn | **88.6** | **78.9** |
| $\times 4$ turn | 88.4 | **78.9** |

## I  EVALUATION ON ADDITIONAL BENCHMARKS

We evaluate ReVeal on two additional code-generation benchmarks: HumanEval+ (Liu et al., 2023) and MBPP+ (Liu et al., 2023). The evaluation process follows the hyperparameter configuration specified in Table 3. We use Pass@1 to measure the success rate of the model's final code solutions. Although these benchmarks differ from our training data, ReVeal still yields larger gains over the baselines as shown in Table 13.

## J  GRADIENT-BASED INCENTIVE ANALYSIS FOR TAPO

ReVeal's multi-turn framework explicitly elicits verification turns in the trajectory, allowing verification tokens to be meaningfully optimized. Within this multi-turn framework, we analyze how TAPO modifies the underlying PPO objective and gradients relative to an outcome-only baseline. The analysis below shows that TAPO's credit assignment strengthens the optimization signal for verification, which helps explain why ReVeal widens the verification–generation (V-G) asymmetry. We then contrast TAPO with a naive token-level implementation that lacks turn-aware credit assignment,

illustrating how such a scheme can suffer from misattributed credit and reward gaming, and how TAPO's design avoids these issues and promotes co-evolution of code and test quality.

**Outcome-only PPO baseline.** Consider an outcome-only PPO baseline that uses only the scalar outcome reward $r_{\text{outcome}}$ from Eq. 1, with the same KL and entropy regularizers as ReVeal. Ignoring these shared regularizers, its objective is

$$J_{\text{out}}(\theta) = \mathbb{E}_{\tau \sim \pi_\theta} \big[ r_{\text{outcome}}(\tau) \big]. \tag{9}$$

With the Monte Carlo return in Eq. 5 and our choice $\gamma = 1$, $\lambda = 1$, the token-level return for all tokens before the end-of-sequence is

$$R_t^{\text{out}} = r_{\text{outcome}}(\tau). \tag{10}$$

Let $V_t^{\text{out}}$ be the critic's estimate trained to regress $R_t^{\text{out}}$, and define the corresponding advantages

$$A_t^{\text{out}} = R_t^{\text{out}} - V_t^{\text{out}}. \tag{11}$$

Denote by $\mathcal{T}_{\text{gen}}$ and $\mathcal{T}_{\text{ver}}$ the token indices belonging to generation and verification turns respectively. Up to the usual PPO clipping and scaling factors, the actor gradients for the two roles can be written as

$$g_{\text{gen}}^{\text{base}}(\theta) \propto \mathbb{E} \Big[ \sum_{t \in \mathcal{T}_{\text{gen}}} A_t^{\text{out}} \nabla_\theta \log \pi_\theta(a_t \mid s_t) \Big], \tag{12}$$

$$g_{\text{ver}}^{\text{base}}(\theta) \propto \mathbb{E} \Big[ \sum_{t \in \mathcal{T}_{\text{ver}}} A_t^{\text{out}} \nabla_\theta \log \pi_\theta(a_t \mid s_t) \Big]. \tag{13}$$

Thus, in the outcome-only baseline both generation and verification tokens are trained only through the same outcome-based advantage.

**TAPO-augmented objective and gradients.** TAPO augments the outcome objective with the generation and verification rewards:

$$J_{\text{TAPO}}(\theta) = \mathbb{E}_{\tau \sim \pi_\theta} \Big[ r_{\text{outcome}}(\tau) + \sum_k r_{\text{gen}}^k(\tau) + \sum_k r_{\text{ver}}^k(\tau) \Big], \tag{14}$$

where $r_{\text{gen}}^k$ is the pass-rate based reward at generation turn $k$ and $r_{\text{ver}}^k$ is the test-quality reward at verification turn $k$. TAPO routes these rewards to tokens through the turn-level return in Eq. 6 and the combined return in Eq. 8. Following Eq. 8, we write the TAPO return and advantages as

$$\tilde{R}_t = R_t + R_t^{\text{turn}}, \qquad A_t^{\text{TAPO}} = \tilde{R}_t - V_t^{\text{TAPO}}, \tag{15}$$

where $V_t^{\text{TAPO}}$ is the critic trained to regress $\tilde{R}_t$. Using Eq. 6 together with the fact that $R_t = r_{\text{outcome}}$ for all tokens before EOS, we obtain explicit forms for the TAPO return. For a token $t$ belonging to generation turn $k$,

$$R_t^{\text{turn}} = r_{\text{gen}}^k, \qquad \tilde{R}_t = r_{\text{outcome}} + r_{\text{gen}}^k. \tag{16}$$

For a token $t$ belonging to verification turn $k$, whose successor turn $k + 1$ is generation, Eq. 6 gives

$$R_t^{\text{turn}} = r_{\text{ver}}^k + r_{\text{gen}}^{k+1}, \qquad \tilde{R}_t = r_{\text{outcome}} + r_{\text{ver}}^k + r_{\text{gen}}^{k+1}. \tag{17}$$

The corresponding actor gradients under TAPO are

$$g_{\text{gen}}^{\text{TAPO}}(\theta) \propto \mathbb{E} \Big[ \sum_{t \in \mathcal{T}_{\text{gen}}} A_t^{\text{TAPO}} \nabla_\theta \log \pi_\theta(a_t \mid s_t) \Big], \tag{18}$$

$$g_{\text{ver}}^{\text{TAPO}}(\theta) \propto \mathbb{E} \Big[ \sum_{t \in \mathcal{T}_{\text{ver}}} A_t^{\text{TAPO}} \nabla_\theta \log \pi_\theta(a_t \mid s_t) \Big]. \tag{19}$$

**Incremental gradients and widened V–G asymmetry.** To make the difference to the outcome-only baseline explicit, we define the incremental gradients

$$\Delta g_{\text{gen}}(\theta) \propto \mathbb{E}\Big[ \sum_{t \in \mathcal{T}_{\text{gen}}} \big(A_t^{\text{TAPO}} - A_t^{\text{out}}\big) \nabla_\theta \log \pi_\theta(a_t \mid s_t)\Big], \tag{20}$$

$$\Delta g_{\text{ver}}(\theta) \propto \mathbb{E}\Big[ \sum_{t \in \mathcal{T}_{\text{ver}}} \big(A_t^{\text{TAPO}} - A_t^{\text{out}}\big) \nabla_\theta \log \pi_\theta(a_t \mid s_t)\Big], \tag{21}$$

so that

$$g_{\text{gen}}^{\text{TAPO}}(\theta) = g_{\text{gen}}^{\text{base}}(\theta) + \Delta g_{\text{gen}}(\theta), \qquad g_{\text{ver}}^{\text{TAPO}}(\theta) = g_{\text{ver}}^{\text{base}}(\theta) + \Delta g_{\text{ver}}(\theta). \tag{22}$$

Using the definitions above we can write

$$A_t^{\text{TAPO}} - A_t^{\text{out}} = \big(\tilde{R}_t - V_t^{\text{TAPO}}\big) - \big(R_t^{\text{out}} - V_t^{\text{out}}\big) = R_t^{\text{turn}} - \big(V_t^{\text{TAPO}} - V_t^{\text{out}}\big). \tag{23}$$

In the actor update, the critic outputs $V_t$ are treated as baselines and are not differentiated with respect to $\theta$. Thus the additional actor gradients $\Delta g_{\text{gen}}$ and $\Delta g_{\text{ver}}$ are driven by the turn-level returns $R_t^{\text{turn}}$ that TAPO introduces.

For generation tokens, $R_t^{\text{turn}} = r_{\text{gen}}^k$ depends only on the per-turn code pass-rate signal and does not involve the verification reward. Relative to outcome-only PPO, TAPO therefore provides finer, outcome-aligned signals for generation by decomposing changes in pass rate into turn-level contributions; this can be viewed as process-level reward shaping rather than introducing a new objective beyond code quality. For verification tokens, in contrast, $R_t^{\text{turn}} = r_{\text{ver}}^k + r_{\text{gen}}^{k+1}$ combines two environment-grounded signals: test quality on the golden code and the effect of these tests on the next generation turn. The outcome-only baseline optimizes verification tokens only through signals mediated by final code quality, whereas TAPO adds verification-specific gradient components that directly reward constructing high-quality, informative tests.

Thus, on the one hand, ReVeal's multi-turn framework explicitly elicits verification turns in the trajectory, allowing verification tokens to be meaningfully optimized, which is not guaranteed by other baseline methods. On the other hand, relative to outcome-only PPO, TAPO strictly enriches the task-aligned reward signals: it explicitly optimizes the verification objective while leaving the generation objective essentially unchanged up to finer credit assignment. This structural asymmetry in the incremental gradients provides a gradient-level explanation for our empirical finding that TAPO substantially increases verification accuracy and widens the V-G gap compared to other baselines.

**Accurate turn-aware credit assignment and robustness to reward gaming.** A naive token-level implementation for multi-turn generation–verification would simply attach each reward $r_{\text{gen}}^k$ or $r_{\text{ver}}^k$ to the last token of the corresponding turn and then use a standard Monte Carlo return for all earlier tokens in the trajectory, without the turn-aware routing in Eq. 6–8. This would propagate the verification reward $r_{\text{ver}}^k$ back through all preceding generation tokens and introduce gradient terms of the form $r_{\text{ver}}^k \nabla_\theta \log \pi_\theta(a_t^{\text{gen}} \mid s_t^{\text{gen}})$ for tokens that do not causally influence $r_{\text{ver}}^k$, since $r_{\text{ver}}^k$ is evaluated on the golden code. Code tokens would then be updated by a signal they do not influence (e.g., wrong code but strong tests still giving high verification reward to the code, or correct code but weak tests inducing low verification reward), and the policy may learn spurious correlations between code patterns and high test accuracy rather than improving true code quality.

By contrast, TAPO's reward design enforces accurate turn-level credit assignment, which is crucial for stable and effective RL training. Its turn-level routing ensures that generation tokens are rewarded only via outcome and pass-rate improvements ($r_{\text{outcome}}$ and $r_{\text{gen}}^k$), and that verification tokens are rewarded only when their tests are both strong on the golden code $r_{\text{ver}}^k$ and helpful for subsequent code refinement $r_{\text{gen}}^{k+1}$ and $r_{\text{outcome}}$. Together with the shared policy for both roles, this structure encourages co-evolution of code and tests: better tests expose more informative failure modes and increase $r_{\text{gen}}^k$ and $r_{\text{outcome}}$, while better code raises the bar for verification and encourages tests that achieve higher $r_{\text{ver}}^k$.

The same credit-assignment pattern naturally extends to more general generation–verification designs with task-specific verification rewards, while remaining robust and compatible across different tasks. For example, if a verification turn were rewarded for judging correctness, the naive implementation

mentioned above could lead to collusion between generation and verification: generation might collapse to trivially wrong solutions that are extremely easy for verification to flag as incorrect, allowing verification to obtain high verification reward while the solution quality remains poor. TAPO's separation of roles prevents such self-referential reward exploitation and make the ReVeal training paradigm robust across different verification reward designs.

## K    LIMITATIONS

Our current instantiation of ReVeal is primarily optimized for functional correctness: solutions that produce correct outputs on the available golden tests are treated as correct, and violations of time/space complexity are only penalized when they cause failures under these tests (e.g., timeouts within the execution budget). As a result, brute-force or computationally inefficient solutions that still pass all golden tests may not be distinguished from truly efficient ones. A natural direction for future work is to add stress tests with very large inputs; these test cases can then be used to compute the outcome and generation rewards, directly encouraging solutions that remain correct under stricter time/space constraints. Another complementary direction is to enrich the tool interface with runtime and resource statistics (e.g., execution time, memory usage) and incorporate these signals into the outcome, generation, and verification rewards, encouraging both efficient code and stress test cases that reliably expose inefficient solutions.

