# OpenReview forum: "ReVeal: Self-Evolving Code Agents via Reliable Self-Verification"
_ICLR.cc/2026/Conference — ICLR 2026 Poster_

### Official Review · Reviewer_6pm9 · 2025-10-27

**Soundness:** 3
**Presentation:** 2
**Contribution:** 2
**Rating:** 2
**Confidence:** 4

**Summary:**

The paper proposes ReVeal, a multi-turn RL method for code generation explicitly optimizing for self-verification, i.e., generating unit test cases in this context, for which the authors adapt PPO into TAPO (Turn-Aware Policy Optimization) through deliberate reward shaping for the "generator" turn and the "verifier" turn. The authors train DAPO-Qwen-32B on a processed subset of TACO and eval on LiveCodeBench (LCB) v6 and CodeContest. The reported Pass@1 = 38.7% on LCB v6 at 25 turns. They also show higher pass@k compared to single-turn RL.

**Strengths:**

About the originality, the problem of getting more signal than existing public test cases is crucial to multi-turn code generation in the competitive programming problems domain. The paper proposes to use model to self-generate test cases, and explicitly optimizes for this in the training time using Reinforcement Learning.

To train a single model end-to-end with the ability of both generating code solutions and unit test cases, the paper details the design of different reward for generation turn and verification turn, which the authors show that it leads to better performance empirically, and its effectiveness is shown via the ablation by comparing to RL training with outcome only reward.

I appreciate the authors' effort on getting results with mean and std deviation with running multiple experiments to reduce the noise of the numbers. Besides reporting the pass rate, the reported metrics of corrections and degradations are also interesting.

**Weaknesses:**

Despite the problem the manuscript tries to tackle and the approach it takes are interesting: the generation-verification gap and the method to train a single model trained for both verification and generation, there're several points that I feel less convincing and the manuscript could be of better form and positioned if these points are properly addressed:

1. Novelty & Contextualization

1.1 I see the core idea of the manuscript as explicitly optimizing self-verification ability, and better turn-level credit assignment, going beyond existing multi-turn RL work RLEF. Therefore, the algorithmic novelty sits largely in the reward shaping design and the TAPO construction, which is consistent with standard PPO variants and MC returns with custom shaping at turn level.

Since the design of TAPO involves multiple component. Ablation of outcome only reward and single-turn RL seems not sufficient for me. These reward shaping necessites better motivation and comparison. The manuscript would benefit from more empirical evidence of the choice of the specific design of reward beyond motivation for justification. To name a few, for example:
- the format reward and the pass rate reward: is the format reward shaping essential and removing it could lead to different performance.
- generation reward: did the author try settings other than `abs = 0` and `imp = 1`?

1.2 Apart from the concrete algorithmic design. The baseline the paper compares to, ReAct is a prompt-only baseline, CTRL is another approach where the verifier is trained against a fixed generator. If the claimed novelty is to explicitly optimize for self-verification, comparing to stronger baseline or alternative design provides stronger evidence of the proposed framework: For example, it's still not clear the gain of the proposed framework comes from the framework design or the RL loss design, which is not totally decoupled.

This is my biggest concern when reading the manuscript: it tries to center around these 2 novelties but provides under-substantial ablation to decouple and justify each of the both. For example, for the framework design thing, the manuscript could compare (under the same classical RL loss such as PPO, or turn-level PPO as shown in the previous work RLEF) the following alternative designs that gradually leads to the current design:

- the verification step is coming from a fixed model, similar to https://arxiv.org/abs/2306.09896 or the mirroring setting of CTRL, either from the same starting policy or from another model, while the generation step is normally RL trained.
- starting from the same starting policy but maintained 2 set of weights separately for generation and verification.
- starting from the same policy, but does 2 separate RL training update for generation and verification where the loss is masked for one another.

For another thing, there're quite a few components deviated from the standard PPO/GRPO practice in the reward design, please see above.

2. Limitation

The propose framework aims at providing more unit test cases to serve as signal in the multi-turn code generation in the competitive programming framework. Wrong code solutions that leads to wrong answer for hidden test cases could benefit from the proposed framework if a verifier produces new test cases. However, another big set of wrong code solutions, which are functionally correct but do not have the desired time/space complexity, could fall out of the catch of the framework: think of a brute force solution where it produces expected output given input. To correct such solution, the verifier needs to have the ability either to produce signal for stats in runtime, or to produce large test cases to hopefully trigger timeout (which might exceed the context length for example). Discussion on whether the framework can properly address such chanllenge or even discussion in a dedicated limitation section is appreciated.

3. Writing and clarity

- It seems that the "tool feedback" refers to solely python execution environment in this paper, which would be good if this could be confirmed.
- The short-term memeory is only briefly mentioned. I thought it means in the multi turns, it only kept recent history (like most recent 3 turns), but L699 says "Critical information such as successful patterns, error types, and effective test structures are preserved in complete formats." Further explanation or example would be appreciated.
- The reward design part necessite more clarity. For example, I'm confused since Verification reward is "For each verification turn k (even), we reward the proportion of generated tests that succeed when executed on a golden code", which therefore, doesn't depend on the generated code but only on the golden code. Which should make it unhackable, but the paper later says in L222 "To mitigate adversarial reward gaming (e.g., generating trivial code that hacks the verification reward) ...". I do not understand how this could happen.
- Eq (5) seems not quite useful as the manuscript never present the token level reward r_t. The whole reward design makes it look like there're token-level return and turn-level return. If I understand correctly, in the end the return is the same for all tokens in the same turn? Is there case where the return is different within one turn for different token? My understanding for the current moment is that the whole design for return is actually at the turn level, this return is "densified" by broadcasting to each token, and the advantage for each token is  this return substract a value baseline.
- L214 says "replaces GAE-based advantages with a turn-aware return." I do not think the author mean it or otherwise the authors use the term "advantage" and "return" interchangeably (which I found confusing), since there're still the advantage estimation in Eq(8), where the V I presume is coming from a value model?

**Questions:**

Please see above for major questions. Also minor question:
- The number on CodeContests is on validation set or test set? I didn't find such information.

---

> ### Author Response · Authors · 2025-11-24
> **Response by authors [1/3]**
>
> We thank the reviewer for the constructive feedback and for recognizing the originality and importance of our approach, as well as our efforts toward a thorough empirical evaluation. We address the concerns below.
>
> ### **Response to W1: Lack of ablations and justification of design choices**
>
> **Response to W1.1: Justification and ablations for reward components**
>
> **(1) Format reward.**
>
> The format reward is essential for ReVeal training. It (i) enforces the prescribed generation-turn / verification-turn tags so we can identify each turn and assign the correct turn-level reward, and (ii) constrains the code and test-case format, especially the test-case input/output blocks, so we can reliably extract code and test cases for tool execution. In our main experiments with DAPO-Qwen-32B, whose instruction-following ability is relatively weak, TAPO only starts to work well once the format is learned reasonably well; when the format score is low, many rollouts are not passable and turn-level rewards cannot be computed. We therefore treat the format reward as a necessary part of the setup and have already clarified its role in the revision.
>
> **(2) Effect of abs/imp coefficients in the generation reward.**
>
> Our generation reward decomposes into an absolute term (abs, code accuracy of the current turn) and an improvement term (imp, how much the current turn’s code accuracy improves over the previous turn), following the **progress-based shaping** used in SCoRe [1] (Section 5.2). The improvement term encourages the model to actually fix failing problems and amplify genuine multi-turn self-correction; SCoRe uses abs = 1, imp = 10. In our setting we use 0/1, i.e., we only use the change in code pass rate to reward the quality of generated tests and the current-turn correction. In early experiments on Qwen2.5-32B-Instruct, we compared several weightings (abs/imp = 0/1, 1/1, 1/0) and observed 0/1 > 1/1 > 1/0, indicating that emphasizing improvement over absolute accuracy leads to better multi-turn performance. We added training and evaluation curves in Appendix G.
>
> **Reference**
>
> [1] Training Language Models to Self-Correct via Reinforcement Learning
>
> **Response to W1.2: Separating framework-level and RL training gains**
>
> We thank the reviewer for pointing out the need to disentangle the contribution of multi-turn framwork from RL training. In the original submission (Appendix C, Table 6), we included a ReAct-style multi-turn prompting baseline (i.e. ReVeal framwork with same prompt and tool) on Qwen2.5-32B-instruct, aimed at disentangling the benefit of the multi-turn framework from that of RL training, results show that framwork only cannot sustain effective deep multi-turn refinement, but ReVeal with RL-trained verification delivers consistent gains. In the revision, we refine the description and consolidate it with new ablations in Appendix F.
>
> **(1) Single-turn RL-trained generator + base model as verifier.**
>
> Following your suggestion, we added a stronger baseline: a fixed verifier (base model) paired with a single-turn RL-trained generator. It brings no improvements due to large $\Delta_{\downarrow}$. Results double confirm that the main gains stem from RL training rather than from the framework alone.
>
> | Configuration                                             | LiveCodeBench V6 Pass@1 ($\Delta_{\uparrow}$, $\Delta_{\downarrow}$)                          |
> |-----------------------------------------------------------|-------------------------------------------------|
> | Single-turn RL model                   |  32.8
> | Single-turn RL model as Generator + Base model as Verifier                   | 33.4@turn1 → 31.5@turn5 ($\Delta_{\uparrow}$=2.36, $\Delta_{\downarrow}$=2.57) → 33.1@turn8($\Delta_{\uparrow}$=1.93, $\Delta_{\downarrow}$=1.50)

---

> > ### Author Response · Authors · 2025-11-24
> > **Response by authors [2/3]**
> >
> > **(2) Two separate policies for generation and verification vs. a shared policy.**
> >
> > We chose a single shared policy for both generation and verification rather than two separate models for two reasons:
> >
> > (i) Using separate models would substantially increase system complexity and RL training/inference cost; as rollouts dominate the cost in RL training and two policies would roughly double it. This is prohibitive for 32B-scale models under our resources, so we apologize that we cannot run this variant during the rebuttal.
> >
> > (ii) Both test generation and code generation require deep code understanding, and prior RL work shows that training on one task can improve out-of-distribution performance on related tasks; with our joint reward design that couples code and test quality, using a single shared policy allows gains in one role (verification or generation) to directly benefit the other.
> >
> > **(iii) Separate RL updates for generation and verification with masked losses for one another.**  We interpret the suggestion of “masking loss for generation or verification” as masking the RL loss for all generation tokens or verification tokens. Taking masking the verification loss as an example, this would remove all learning signal for verification, including the format, and would also hurt the optimization of later generation turns because they would receive poor feedback from the verification. Given this and our limited compute budget, we instead provide a lower-cost and stronger variants, ReVeal (no-ver RL) and ReVeal (no-gen RL) , where we apply the ReVeal multi-turn framework, use the ReVeal-trained model for one role (e.g., generater) and untrained base model for the other role (e.g., verifier). The results show that ReVeal (no-ver RL) yields gains in the first 5 turns and quickly saturates, whereas the full ReVeal model continues to improve with more turns with higher $\Delta_{\uparrow}$ and lower $\Delta_{\downarrow}$, indicating that ReVeal’s explicit optimization of verification yields clear performance gains and enables deeper test-time scaling. ReVeal (no-gen RL) continues to improve with more turns even when paired with a relatively weak generator, indicating that the deeper test time scaling indeed come from ReVeal’s training of the verification tokens.
> >
> > | Configuration                                              | LiveCodeBench V6 Pass@1 ($\Delta_{\uparrow}$, $\Delta_{\downarrow}$)                                          |
> > |-----------------------------------------------------------|----------------------------------------------------------------------------------------------------------------|
> > | ReVeal (no-gen RL): Base model as Generator + ReVeal as Verifier | 29.0@turn1 → 33.6@turn5 ($\Delta_{\uparrow}$=6.16, $\Delta_{\downarrow}$=0.91) → 33.8@turn8 ($\Delta_{\uparrow}$=6.39, $\Delta_{\downarrow}$=0.91) |
> > | ReVeal (no-ver RL): ReVeal as Generator + Base model as Verifier | 34.6@turn1 → 36.9@turn5 ($\Delta_{\uparrow}$=2.94, $\Delta_{\downarrow}$=0.53) → 36.8@turn8 ($\Delta_{\uparrow}$=4.25, $\Delta_{\downarrow}$=1.26) |
> > | ReVeal (full RL)                                          | 34.8@turn1 → 37.2@turn5 ($\Delta_{\uparrow}$=3.71, $\Delta_{\downarrow}$=0.0) → 37.7@turn8 ($\Delta_{\uparrow}$=5.62, $\Delta_{\downarrow}$=0.0)   |
> >
> >
> > ### **Response to W2: Limitation on time/space complexity**
> >
> > We agree that our current framework mainly optimizes functional correctness: solutions that are functionally correct but have suboptimal time/space complexity are only penalized when they fail on the available golden tests.
> >
> > A natural way to address this in future work is to strengthen the training test sets with more stress tests. For example, one can automatically generate large inputs by code, run them on a golden solution offline to obtain outputs, and then use these large test cases when computing the outcome reward and the generation reward; in this setting, our rewards would directly encourage code that handles such inputs within the time/space limits during training. In addition, the tool interface could expose runtime or resource statistics so that these signals can be incorporated into the outcome reward and the generation reward, further encouraging efficient code beyond pure functional correctness, and into the verification reward to encourage generating stress test cases. Letting the LLM itself generate extremely large inputs that exceed the context window, and reason about the corresponding large outputs without access to the golden code, would remain very challenging (even harder than generating code) and is not the focus of this work; we've added discussion in Appendix K.

---

> > > ### Author Response · Authors · 2025-11-24
> > > **Response by authors [3/3]**
> > >
> > > ### **Response to W3: Writing and clarity**
> > >
> > > **(1) Clarification on tool feedback**: We confirm that it refers solely to the Python code-execution environment.
> > >
> > > **(2) Clarification on short-term memory**: We confirm that at inference time we keep only the last three generation–verification turns as context. Within those three turns, we preserve the full content of code, tests, and tool feedback. We revised it with a more precise description in Appendix C.
> > >
> > > **(3) Clarification on reward design**: We confirm that the verification reward is the proportion of generated tests that pass on the golden code in our setting. TAPO’s reward design enforces **accurate turn-level credit assignment**, which is crucial for stable and effective RL training. If we naively propagated the verification reward to all preceding tokens, code tokens would be updated by a signal they do not causally influence (e.g., wrong code but good tests still giving high verification reward to code, or correct code but weak tests inducing low verification reward), and the policy may **learn spurious correlations between code patterns and high test accuracy** rather than improving true code quality.
> > >
> > > Our design is deliberately formulated for general generation–verification setting, remaining robust and compatible with different verification reward designs across tasks, so that the ReVeal RL training paradigm can be applied whenever a generation turn proposes solutions and a verification turn evaluates them. For example, if the verification turn were rewarded for judging correctness, then generation and verification could collude: generation can collapse to trivially wrong solutions that are extremely easy for verification to flag as incorrect, allowing verification to obtain high “verification” reward while the solution quality remains poor (Please see Appendix J for details).
> > >
> > >
> > > **(4) Clarification on Eq. (5) and token- vs. turn-level returns**: We apologize that the original manuscript did not explicitly define the token-level reward $r_t$, and we have clarified this in the revised version. The token-level rewards $r_t$ are defined as $r_T = r_{\text{outcome}}$ in Eq. (1) for the last token $T$ of the response, and $r_t = 0$ for all $t < T$. Eq. (5) then computes the **standard Monte Carlo return from these $r_t$** (with $\gamma = 1, \lambda = 1$), so this **token-level return assigns the same outcome reward to all tokens in the trajectory**, exactly as in standard PPO with a single outcome reward.
> > >
> > > In contrast, the **turn-aware component is driven by the generation and verification rewards**: turn-level rewards are attached to the last token of each turn, and the resulting turn-level return is assigned only to the tokens specified by Eqs. (6–7), rather than to all preceding tokens. Eq. (8) then combines the outcome-based token-level return and the turn-level shaping into a single effective return. As the reviewer inferred, in our design there is no case where different tokens within the same turn receive different returns, and advantages are computed as this combined return minus the value baseline.
> > >
> > > **(5) Clarification on wording around GAE**: Thank you for pointing out this ambiguity. In the revision, we clarify this by rewriting the sentence at L218 as: TAPO only modifies the advantage estimator: instead of using the standard GAE-based advantages, it uses a turn-aware return to construct the advantage estimates.
> > >
> > > ### **Response to Q1: CodeContests evaluation set**
> > >
> > > We used CodeContests test set for evaluation.

---

> > > > ### Comment · Reviewer_6pm9 · 2025-11-27
> > > > **Comment to the rebuttal**
> > > >
> > > > I thank the authors for the explanation and the efforts to put into the rebuttal!
> > > >
> > > > As I noted in the original review, in my point of view, the main contributions of the paper are:
> > > >
> > > > 1. the framework design that brings the generator & verifier framework to multi-turn RL training time
> > > > 2. the proposed TAPO algorithm for end-to-end training generator and verifier in the same policy and better turn-level credit assignment
> > > >
> > > > Also, as I noted in the strength, the Delta metrics is also interesting and provides hints beyond just reporting pass rate. And I highly appreciate the authors' effort on getting results with mean and std deviation with running multiple experiments.
> > > >
> > > > *These 2 contributions are novel, timely and address one of the main bottlenecks of the current frontier of competitive programming in the multi-turn settings. That being said, if these 2 claims/contributions are extensively empirical justified, this manuscript would be a solid contribution.*
> > > >
> > > > This is also my concern in my original review: I did think we need more empirical evidence to justify the source of gain, and decouple the contribution 1.& 2. to provide clear insight on how they impact the final performance. Reviewer utHn and Reviewer beFR also raised the same question. My comments to the rebuttal are below:
> > > >
> > > > 1. The reward component
> > > >
> > > > Thank you for the additional plot and insight for the abs/imp variants. This shows how much gain we can get compared to the vanilla pass rate as the reward setting.
> > > >
> > > > 2. The Single-turn RL-trained generator + base model as verifier and The ReVeal (no-gen RL), ReVeal (no-ver RL) and ReVeal (full RL) comparison.
> > > >
> > > > Thank you for proposing this suite of comparison! I agree that this is cleaner than my original proposed exp. in my comment. The 2nd set of experiments clearly show how much gain we can have by bring the verifier to training time as well, compared to a fixed verifier. With comparing "Single-turn RL model as Generator + Base model as Verifier" and "ReVeal (no-ver RL)", we can also know the gain of doing the verifier framework in test time only and in training time, given that in both setting the verifier is fixed.
> > > >
> > > > In all, I think the authors nicely address my main concern about decoupling the 2 main contributions stated in the paper. I will adjust my score to reflect the improvement.
> > > >
> > > > Still, I find that my confusion is not fully resolved regarding the reward design part and maybe some misunderstanding still exists. I would appreciate if the authors could help me understand the following question for clarification:
> > > >
> > > > 1. For the token-level return, it's the cumulative reward for the $r_{outcome}$, which is $r_{format} + r_{passrate}$, do I understand correctly that this token-level return only apply on tokens from the generator, but not the tokens from the verifier? Since if I read Eq (5), the sum is from token $t$ to the end of the trajectories, which contains both generator tokens and verifier tokens.
> > > >
> > > > 2. Thanks to the authors' explanation I understand that the verifier's reward should not propagate to all preceding tokens (probably due to the return calculation?). "turn-level rewards are attached to the last token of each turn, and the resulting turn-level return is assigned only to the tokens specified by Eqs. (6–7)" I actually have confusion here, could the authors help me with this example?
> > > >
> > > > Consider a token sequence [g0, g1, g2, g3, g4, v0, v1, v2, v3, v4, g5, g6, g7, g8, g9], where g stands for generator and v stands for verifier. The paper uses "{t_1, . . . , t_K}" to denotes the token indices of the end of turn. In this case it would be {g4, v4, g9}.
> > > >
> > > > From Eq(6), $R^{turn}(g4)$ would be the generator reward at g4, $R^{turn}(g9)$ at g9 respetively, $R^{turn}(v4)$ would be the verifier reward + $R^{turn}(g9)$. All good.
> > > >
> > > > Proceed to Eq(7), for token $t$, define $\tau(t) = \min \{ t_k | t_k \geq t \} $, so for g0 to g4, this quantity is g4, for v0 to v4, this quantity is v4 etc, and $R^{turn}_t$ = $R^{turn}(\tau(t))$. Therefore, for g0 to g4, $R^{turn}_t$ are all generator reward at g4, same for v0 to v4 and same for g5 to g9: this just copy the end-of-turn return to every token.
> > > >
> > > > Therefore the "turn-level return" restrict to current turn (for generator) and deviate from the term in classical RL sense which stands for cumulative backward sum?

---

> > > > > ### Author Response · Authors · 2025-11-28
> > > > >
> > > > > Thank you again for the careful follow-up and for updating your score, as well as for your encouraging comment that, with stronger empirical justification, our framework and TAPO could become a solid contribution.
> > > > >
> > > > > For your first question about the token-level return in Eq. (5): in our implementation the outcome-based return is applied to **all optimized tokens** in the trajectory, including both generation and verification tokens (we only mask out \<tool-feedback\> tokens from the loss). Concretely, we place the outcome reward on the last token of the trajectory and set all intermediate per-token rewards to zero, so the Monte Carlo return is the same outcome value for every token, exactly as in standard outcome-only RLVR. Intuitively, this global outcome return encourages the whole trajectory to learn the required format (generation / verification tags and code / test-case structure) and keeps both roles aligned with final code accuracy, while the turn-level shaping focuses on more local, role-specific signals.
> > > > >
> > > > > For your second question about the turn-level return in Eqs. (6–7) and the toy example: your understanding is correct that all tokens within the same turn receive the same turn-level return. We intentionally treat each turn as a single decision unit. The key difference from a classical cumulative backward sum lies in how returns are propagated **across turns**. Generation turns only receive their own generation reward (plus the global outcome return), so generation tokens are not directly shaped by future verification rewards they do not causally control. Verification turns, on the other hand, receive both their own verification reward and the reward of the next generation turn, so they are explicitly credited when their tests lead to better subsequent code. In this sense, the “turn-level return” is a structured, role-aware shaping term, not a generic cumulative sum over all future rewards as in a standard Monte Carlo return.
> > > > >
> > > > > We will clarify these points in the revised version and are happy to answer any further questions.

---

### Official Review · Reviewer_beFR · 2025-10-30

**Soundness:** 3
**Presentation:** 3
**Contribution:** 3
**Rating:** 4
**Confidence:** 3

**Summary:**

This paper introduces ReVeal, a multi-turn reinforcement learning framework for code generation that explicitly optimizes self-verification alongside code generation. The key innovation is treating verification as an optimization target rather than relying solely on outcome rewards. ReVeal structures long-horizon reasoning as iterative generation-verification loops, where the model generates code, constructs test cases, executes them via external tools (Python interpreter), and uses the feedback to refine solutions across multiple turns. The authors propose Turn-Aware Policy Optimization (TAPO), which assigns credit at both token and turn granularity using joint verifiable rewards (outcome, generation, and verification). Experiments on LiveCodeBench and CodeContests show that ReVeal enables test-time scaling beyond the training horizon and achieves higher Pass@k than baseline methods, suggesting expanded reasoning boundaries.

**Strengths:**

The paper presents a genuinely novel perspective on multi-turn code generation by explicitly optimizing verification as a co-equal objective with generation. While prior work has explored critic models or execution feedback, ReVeal's approach of jointly training generation and verification within a single model through structured turn-level rewards is innovative. The TAPO algorithm, though building on PPO, introduces a sensible credit assignment mechanism tailored to the generation-verification paradigm.

The paper is generally well-structured with clear motivation. Figure 3 effectively illustrates the generation-verification loop and TAPO's reward structure. The case study provides intuitive understanding of how the iterative refinement process works in practice. The distinction between training (with golden solution filtering) and inference (fully autonomous) is clearly articulated.

**Weaknesses:**

The ablation analysis is concerningly limited for a paper making multiple methodological contributions. Table 1 only compares "outcome reward" versus "TAPO with joint rewards" as a monolithic change, without isolating the contribution of individual components. Critical missing ablations include: (1) What happens with only generation rewards or only verification rewards? (2) How does the specific turn-level return formulation in the equation compare to simpler alternatives? (3) What is the effect of different reward hyperparameters (abs/imp coefficients, the factor of 5 in passrate)? (4) Why is the ablation only shown at turn 8 when main results extend to turn 25? These gaps make it difficult to assess which design choices truly matter.

The paper claims to "widen the verification-generation asymmetry" (Figure 2) but provides only empirical observation without theoretical grounding. Why does TAPO's specific formulation in Equations 6-8 prevent reward hacking? The intuitive explanation that "generation turns are rewarded solely based on code quality" is not rigorously proven. What are TAPO's convergence properties compared to standard PPO? The paper would benefit from formal analysis showing that TAPO's credit assignment scheme provably incentivizes the desired co-evolution behavior rather than reward exploitation.

Despite claiming the method is "applicable to any reasoning task with verifiable rewards" and "verification asymmetry," experiments are confined entirely to code generation on two similar benchmarks (LiveCodeBench, CodeContests). No evidence supports the broader applicability claim. Even within code generation, evaluation is limited to competitive programming problems—what about software engineering tasks like debugging, refactoring, or API usage? The paper tests only two base models from the same model family (Qwen). Testing on diverse model architectures and problem types would strengthen generalization claims significantly.

**Questions:**

Can you provide complete ablations isolating: (a) generation reward only, (b) verification reward only, (c) outcome + generation, (d) outcome + verification? This would clarify each component's contribution. Additionally, what happens with different reward hyperparameters (the factor of 5, abs/imp coefficients)?

What percentage of model-generated tests are filtered during training because they fail on the golden solution? How does this evolve over training? Since filtered tests provide no learning signal, how does the model learn to avoid generating bad tests?

Can you provide theoretical or empirical analysis of TAPO's convergence properties? How does the combination of token-level and turn-level returns (Equation 8) affect optimization stability compared to standard GAE? Have you observed any instabilities?

---

> ### Author Response · Authors · 2025-11-24
> **Response by authors [1/5]**
>
> We thank the reviewer for the constructive feedback and for recognizing that ReVeal presents a genuinely novel perspective by explicitly optimizing verification as a co-equal objective with generation, together with a sensible turn-level credit assignment mechanism and a clearly motivation and framework. We address the concerns below.
>
> ### **Response to W1: Lack of ablations and analysis of design choices**
>
> **(1)  Ablations on generation and verification rewards**
>
> Following the reviewer’s suggestion, we add two ReVeal variants in addition to the ReVeal outcome-only baseline: ReVeal outcome+gen and ReVeal outcome+ver. As shown in the table below, both variants outperform the outcome-only version, and the full ReVeal achieves the overall best results. This indicates that both $r_{\text{gen}}$ and $r_{\text{ver}}$ make useful contributions beyond the outcome reward, with $r_{\text{gen}}$ supervises both generation and verification by encouraging exposing more informative failure modes and larger subsequent code improvements, while $r_{\text{ver}}$ supervises test case accuracy directly.
>
> | Model                         | LiveCodeBench V6 Pass@1                     | CodeContests Pass@1                        |
> |-------------------------------|----------------------------------------------|--------------------------------------------|
> | ReVeal (outcome only)         | 32.7@turn1 → 35.9@turn5 → 36.1@turn8        | 20.0@turn1 → 26.1@turn5 → 27.4@turn8       |
> | ReVeal (outcome+ver)          | 32.3@turn1 → 36.6@turn5 → 36.8@turn8        | 17.3@turn1 → 26.5@turn5 → 27.8@turn8       |
> | ReVeal (outcome+gen)          | 30.7@turn1 → 37.1@turn5 → 38.0@turn8    | 17.2@turn1 → 26.6@turn5 → 28.2@turn8       |
> | ReVeal Full (outcome+gen+ver) | 34.8@turn1 → 37.2@turn5 → 37.7@turn8        | 22.3@turn1 → 28.1@turn5 → 30.4@turn8   |
>
> Regarding variants with only generation or verification rewards without the outcome reward, we view the outcome reward as essential, since it is the only signal that directly optimizes the final evaluation metric (final code quality). All RL baselines in Table 1 are trained with this objective; training variants that omit it would be artificially weakened and make the comparison to these baselines less meaningful. Given the high cost of 32B-scale RL, we focus ablations on practically relevant settings that retain the outcome reward.
>
> **(2) Turn-level return formulation vs simpler alternatives**
>
> By “simpler alternatives”, we understand this as using standard PPO with $r_{\text{gen}}$ and $r_{\text{ver}}$ placed on the last token of each turn and propagating them to all preceding tokens, without the turn-aware shaping in Eqs. (6–8). We implemented such PPO-style variants in early experiments, but these attempts failed: training was noticeably less stable and converged to worse performance. We attribute this to **misaligned credit assignment**: code tokens are updated by a signal they do not causally influence (e.g., wrong code but good tests still giving high verification reward to code, or correct code,  but weak tests inducing low verification reward), and the policy may learn spurious correlations between code patterns and high test accuracy rather than improving true code quality.
>
> TAPO was introduced to address this by enforcing precise turn-level credit assignment, which in our experiments yielded more stable training and better final performance. We will add a short discussion of these simpler PPO variants in the appendix J.

---

> > ### Author Response · Authors · 2025-11-24
> > **Response by authors [2/5]**
> >
> > **(3) Reward hyperparameters**
> >
> > **Effect of abs/imp coefficients in the generation reward.**
> > Our generation reward decomposes into an absolute term (abs, code accuracy of the current turn) and an improvement term (imp, how much the current turn’s code accuracy improves over the previous turn), following the **progress-based shaping** used in SCoRe [1] (Section 5.2). The improvement term encourages the model to actually fix failing problems and amplify genuine multi-turn self-correction; SCoRe uses abs = 1, imp = 10. In our setting we use 0/1, i.e., we only use the change in code pass rate to reward the quality of generated tests and the current-turn correction. In early experiments on Qwen2.5-32B-Instruct, we compared several weightings (abs/imp = 0/1, 1/1, 1/0) and observed 0/1 > 1/1 > 1/0, indicating that emphasizing improvement over absolute accuracy leads to better multi-turn performance. We added training and evaluation curves in Appendix G.
> >
> > **Effect of the “factor of 5” in pass-rate.**  The “factor of 5” is an empirical balancing coefficient between the format reward and the pass-rate reward. Unlike some work that drop the format term[1,2], in our setting we rely on the format reward to enforce a strict, parsable format for test-case and turn tags, instead we reduce its relative weight. Without scaling, early in training the model tends to satisfy the format constraints more quickly, so gradients would be dominated by format rather than by code quality. Multiplying the pass-rate term by 5 gives pass-rate a stronger influence on learning. In practice, the format reward is easier to learn and quickly saturates close to 1, after which most useful learning signal comes from pass-rate, so moderate changes to this coefficient do not qualitatively change behavior. Given the high cost of 32B-scale RL, we did not perform an extensive sweep over this coefficient and instead followed prior RL practice.
> >
> > **(4) Explanation about different turns (8 vs. 25) is used for evaluation.**
> >
> > We apologize for the confusion. In the main results, our goal is to show the extreme long-horizon scaling capability of ReVeal, so for the main model (TAPO with joint rewards) we report performance to 25 turns, and Figure 1(a) shows the full Pass@1 curve across all turns, demonstrating continued improvement well beyond the 3-turn training horizon.
> >
> > By contrast, the ablations are meant to compare different reward designs (e.g., outcome-only vs. the joint-reward), not to push each variant to longer inference horizons. Evaluating ablation variants up to 8 turns already covers the regime where multi-turn refinement is active (roughly 2–3× the training horizon) while keeping experiments efficient for 32B models. In the revision, we report detailed results at 1/5/8 turns (see above Table in response to W1), where ReVeal with TAPO (i.e., the main model) consistently outperforms the outcome-only baseline.
> >
> > **Reference**
> >
> > [1] Open-Reasoner-Zero: An Open Source Approach to Scaling Up Reinforcement Learning on the Base Model
> > [2] SimpleRL-Zoo: Investigating and Taming Zero Reinforcement Learning for Open Base Models in the Wild

---

> > > ### Author Response · Authors · 2025-11-24
> > > **Response by authors [3/5]**
> > >
> > > ### **Response to W2: Clarification on widening the V-G asymmetry, reward hacking, and convergence**
> > > We thank the reviewer for the insightful comments. Our work’s main contributions are algorithmic and empirical rather than theoretical, but we agree that a brief formal analysis is useful. In the revised version, we add Appendix J and summarize the main points below.
> > >
> > > **(1) On widening the V–G asymmetry.**
> > >
> > > First, **ReVeal’s multi-turn framework explicitly elicits verification turns in the trajectory, allowing verification tokens to be meaningfully optimized, which is not guaranteed by other baseline methods**. Within this framework, we now add a lightweight incentive-level analysis comparing TAPO to outcome-only PPO, clarifying why our objective places relatively more optimization pressure on verification than outcome-only PPO.
> > >
> > > In the outcome-only baseline, we optimize only the outcome reward $r_{\text{outcome}}$, with the same KL and entropy regularizers as in ReVeal. With $\gamma = 1$ and $\lambda = 1$, the token-level return is constant along the trajectory,
> > >
> > > $$
> > > R^{\text{out}}\_t = r\_{\text{outcome}}(\tau),
> > > $$
> > >
> > > and the advantage is
> > > $$
> > > A^{\text{out}}_t = R^{\text{out}}_t - V^{\text{out}}_t.
> > > $$
> > >
> > > Denoting by $\mathcal{T}\_{\text{gen}}$ and $\mathcal{T}\_{\text{ver}}$ the generation and verification tokens, the actor gradients are
> > >
> > > $$
> > > g^{\text{base}}\_{\text{gen}}(\theta) \propto
> > > \mathbb{E}\Bigg[\sum\_{t \in \mathcal{T}\_{\text{gen}}}
> > > A^{\text{out}}\_t \nabla\_\theta \log \pi\_\theta(a\_t \mid s\_t)\Bigg],\quad
> > > g^{\text{base}}\_{\text{ver}}(\theta) \propto
> > > \mathbb{E}\Bigg[\sum\_{t \in \mathcal{T}\_{\text{ver}}}
> > > A^{\text{out}}\_t \nabla\_\theta \log \pi\_\theta(a\_t \mid s\_t)\Bigg].
> > > $$
> > >
> > > Thus, in the baseline both roles are trained only via the same outcome-based advantage.
> > >
> > > TAPO augments this objective with per-turn generation and verification rewards and routes them through the turn-level return (Eqs. 6–8). In Appendix J we denote the combined return by
> > >
> > > $$
> > > \tilde R\_t = R\_t + R^{\text{turn}}\_t,\quad
> > > A^{\text{TAPO}}\_t = \tilde R\_t - V^{\text{TAPO}}\_t,
> > > $$
> > >
> > > and show that, for a generation token in generation turn \(k\),
> > >
> > > $$
> > > R^{\text{turn}}\_t = r\_{\text{gen}}^k,\quad
> > > \tilde R\_t = r\_{\text{outcome}} + r\_{\text{gen}}^k,
> > > $$
> > >
> > > while for a verification token in verification turn \(k\) followed by generation turn \(k+1\),
> > >
> > > $$
> > > R^{\text{turn}}\_t = r\_{\text{ver}}^k + r\_{\text{gen}}^{k+1},\quad
> > > \tilde R\_t = r\_{\text{outcome}} + r\_{\text{ver}}^k + r\_{\text{gen}}^{k+1}.
> > > $$
> > >
> > > This yields TAPO actor gradients
> > >
> > > $$
> > > g^{\text{TAPO}}\_{\text{gen}}(\theta)
> > > = g^{\text{base}}\_{\text{gen}}(\theta) + \Delta g\_{\text{gen}}(\theta),\quad
> > > g^{\text{TAPO}}\_{\text{ver}}(\theta)
> > > = g^{\text{base}}\_{\text{ver}}(\theta) + \Delta g\_{\text{ver}}(\theta),
> > > $$
> > >
> > > where the incremental gradients are
> > >
> > > $$
> > > \Delta g\_{\text{gen}}(\theta) \propto
> > > \mathbb{E}\Bigg[\sum\_{t \in \mathcal{T}\_{\text{gen}}}
> > > \big(A^{\text{TAPO}}\_t - A^{\text{out}}\_t\big)\,
> > > \nabla\_\theta \log \pi\_\theta(a\_t \mid s\_t)\Bigg],
> > > $$
> > >
> > > $$
> > > \Delta g\_{\text{ver}}(\theta) \propto
> > > \mathbb{E}\Bigg[\sum\_{t \in \mathcal{T}\_{\text{ver}}}
> > > \big(A^{\text{TAPO}}\_t - A^{\text{out}}\_t\big)\,
> > > \nabla\_\theta \log \pi\_\theta(a\_t \mid s\_t)\Bigg],
> > > $$
> > >
> > > and
> > >
> > > $$
> > > A^{\text{TAPO}}\_t - A^{\text{out}}\_t
> > > = (\tilde R\_t - V^{\text{TAPO}}\_t) - (R^{\text{out}}\_t - V^{\text{out}}\_t)
> > > = R^{\text{turn}}\_t - (V^{\text{TAPO}}\_t - V^{\text{out}}\_t).
> > > $$
> > >
> > > In the actor update, the critic outputs $V\_t$ are treated as baselines and are not differentiated with respect to $\theta$. Thus the additional actor gradients $\Delta g\_{\text{gen}}$ and $\Delta g\_{\text{ver}}$ are **driven by the turn-level returns $R^{\text{turn}}\_t$** that TAPO introduces.
> > >
> > > For **generation tokens**, $R^{\text{turn}}\_t = r\_{\text{gen}}^k$ depends only on the per-turn code pass-rate signal and does not involve the verification reward. Relative to outcome-only PPO, TAPO therefore provides finer, outcome-aligned signals for generation by decomposing changes in pass rate into turn-level contributions; this can be viewed as process-level reward shaping rather than introducing a new objective beyond code quality.
> > >
> > > For **verification tokens**, in contrast, $R^{\text{turn}}\_t = r\_{\text{ver}}^k + r\_{\text{gen}}^{k+1}$ combines two environment-grounded signals: test quality on the golden code ($r\_{\text{ver}}^k$) and the effect of these tests on the next generation step ($r\_{\text{gen}}^{k+1}$). **The outcome-only baseline updates verification only through signals mediated by final code correctness, whereas TAPO adds verification-specific gradient components that directly reward constructing high-quality, informative tests.**

---

> > > > ### Author Response · Authors · 2025-11-24
> > > > **Response by authors [4/5]**
> > > >
> > > > This analysis shows that **TAPO strictly enriches the task-aligned reward signals that drive verification updates**, while leaving the generation objective essentially unchanged up to finer outcome-aligned shaping. We view “widening the V–G asymmetry” as precisely this algorithmic bias: verification receives additional task-aligned gradient sources ($r\_{\text{ver}}^k$ and future $r\_{\text{gen}}^{k+1}$) that are absent in the outcome-only baseline. This gradient-level picture is consistent with the behavior illustrated in Figure 2 and with our training curves.
> > > >
> > > > **(2)  "generation turns are rewarded solely based on code quality" is not rigorously proven**
> > > >
> > > > From the gradient expressions above, we see that for generation tokens the **additional gradients introduced by TAPO** depend only on the turn-level return $R^{\text{turn}}\_t = r\_{\text{gen}}^k$; **generation tokens never receive gradients from the verification reward $r\_{\text{ver}}^k$**. In this precise sense, generation turns are updated solely based on code-quality signals.
> > > >
> > > > In contrast, a naive Monte Carlo implementation that attaches $r\_{\text{gen}}^k$ or $r\_{\text{ver}}^k$ to the last token of each turn and backs it up through all earlier tokens would propagate the verification reward $r\_{\text{ver}}^k$ back through all preceding generation tokens and introduce gradient terms of the form $$ r\_{\text{ver}}^k \nabla\_\theta \log \pi\_\theta(a^{\text{gen}}\_t \mid s^{\text{gen}}\_t) $$ for tokens that do not causally influence $r\_{\text{ver}}^k$, since $r\_{\text{ver}}^k$ is evaluated on the golden code. Code tokens would then be updated by a signal they do not influence (e.g., wrong code but strong tests still giving high verification reward to the code, or correct code but weak tests inducing low verification reward), and the policy may learn spurious correlations between code patterns and high test accuracy rather than improving true code quality. Under TAPO, this reward-hacking channel is removed by design.
> > > >
> > > > **(3) Co-evolution**
> > > >
> > > > From the above analysis, we can see that in TAPO, generation tokens are rewarded only via outcome and pass-rate improvements ($r\_{\text{outcome}}$ and $r\_{\text{gen}}^k$), and verification tokens are rewarded only when their tests are both strong on the golden code ($r_{\text{ver}}^k$) and helpful for subsequent code refinement ($r\_{\text{gen}}^{k+1}$) and $r\_{\text{outcome}}$. Together with the shared policy for both roles, this structure encourages co-evolution of code and tests: better tests expose more informative failure modes and increase $r\_{\text{gen}}^k$ and $r\_{\text{outcome}}$, while better code raises the bar for verification and encourages tests that achieve higher $r\_{\text{ver}}^k$.
> > > >
> > > > **(4) Convergence properties of TAPO vs. PPO**
> > > >
> > > > In all our experiments we keep the PPO optimizer and actor–critic architecture unchanged and only modify the reward signal. Concretely, we augment the outcome reward with generation and verification rewards and define a turn-aware per-step reward $r'(s\_t,a\_t)$, which induces the turn-aware return in Eqs. 5–8. The PPO clipped surrogate objective and KL penalty remain the same, so TAPO is simply standard PPO applied to the modified reward function, with underlying objective
> > > >
> > > > $$
> > > > J\_{\text{TAPO}}(\theta)
> > > > = \mathbb{E}\_{\tau\sim\pi\_\theta}\Big[\sum\_t r'(s\_t,a\_t)\Big].
> > > > $$
> > > >
> > > > Under the usual assumptions of policy-gradient theory (finite MDP and bounded rewards), TAPO therefore inherits the standard convergence guarantees of PPO for this objective.
> > > >
> > > > Empirically, TAPO does not introduce additional instability; on the contrary, it avoids the instability of a naive Monte Carlo implementation without Eqs. 6–8. The training curves in Figures 4 and 5 show that TAPO trains stably and does not introduce additional instability compared to the baseline, with very similar convergence behavior.

---

> > > > > ### Author Response · Authors · 2025-11-24
> > > > > **Response by authors [5/5]**
> > > > >
> > > > > ### **Response to W3: Generalization**
> > > > >
> > > > > **(1) Generalization to other tasks.**
> > > > >
> > > > > Our intention is to state that the *ReVeal training paradigm* (multi-turn generation–verification + TAPO credit assignment) is not specific to competitive code task: whenever there exists a verifiable signal for candidate solutions, one can in principle instantiate ReVeal by (i) defining an appropriate verification reward, (ii) wiring this reward into the TAPO return, and (iii) designing task-specific tools and feedback.
> > > > >
> > > > > In the revision, we have added experiments on HumanEval+ and MBPP+, using the same ReVeal-trained models and baselines. Although these benchmarks differ from our training data, ReVeal still yields larger gains over the baselines, suggesting that the benefits of ReVeal are not limited to competition-style problems (Appendix I). For applying ReVeal to debugging, refactoring, or API-usage scenarios would require building suitable test tool or API sandboxes, designing verification rewards which need substantial work. We agree that these software-engineering tasks are highly valuable application domains, and we will clarify them as promising directions for future work.
> > > > >
> > > > > | Model              | HumanEval+ Pass@1 | MBPP+ Pass@1 |
> > > > > |--------------------|-------------------|--------------|
> > > > > | Qwen2.5-32B-Instruct | 85.4             | 75.4         |
> > > > > | DAPO-Qwen2.5-32B     | 86.0             | 73.8         |
> > > > > | Single-turn RL       | 86.0             | 76.2         |
> > > > > | ReVeal×8 (1 turn)    | 85.2             | 77.9         |
> > > > > | ReVeal×8 (3 turns)   | **88.6**         | **78.9**     |
> > > > > | ReVeal×8 (4 turns)   | 88.4             | **78.9**     |
> > > > >
> > > > >
> > > > > **(2) Generalization to other models.**
> > > > >
> > > > > To assess applicability across models under a tight compute budget, we also experimented with smaller models. With Llama-3.2-3B-Instruct, the generation-think part was often empty or extremely short and the verification-think part rarely produced meaningful test-generation reasoning, which indicate that the model is struggling to learn the required format because of weak instruction following.
> > > > >
> > > > > We therefore switched to Qwen3-4B-Instruct-2507. The full results are reported in Appendix E, Table 9.  Resutls show ReVeal on this model still significantly outperforms baseline methods. This supports that our training recipe is not tied to a single scale (32B) and can generalize across different models. Extending to more diverse architectures is interesting future work.
> > > > >
> > > > > | Model                    | LiveCodeBench V6 Pass@1                  | CodeContests Pass@1                     |
> > > > > |--------------------------|-------------------------------------------|-----------------------------------------|
> > > > > | Qwen3-4B-Instruct        | 33.1                                     | 24.3                                    |
> > > > > | Single-turn RL           | 39.0                                     | 26.9
> > > > > | ReVeal| 40.6@turn1 → 44.1@turn5 → **44.5**@turn8 | 28.5@turn1 → 33.6@turn5 → **33.9**@turn8 |
> > > > >
> > > > >
> > > > > ### **Response to Q1: ablation on reward**
> > > > >
> > > > > Please see response to W1 (1).
> > > > >
> > > > > ### **Response to Q2: filtered tests during training and learning signal.**
> > > > > Figure 5(d) reports test accuracy on the golden solution calculated by Eq. 3, so $1-r\_{\text{ver}}^k$  is exactly the fraction of generated tests that fail on the golden code. From the figure, this failure rate decreases significantly over training, showing that the model learns to generate far fewer wrong tests.
> > > > >
> > > > > The question assumes that filtered tests provide no learning signal, which is a misunderstanding. Wrong tests are only filtered **in the tool feedback** (we do not run them against candidate code to get accurate feedback to guild better exploration), but **all generated tests remain in the RL trajectory** and enter the verification reward: failing tests contribute 0 to the pass-rate and therefore lower the verification reward and the corresponding advantages. Thus bad tests are explicitly penalized rather than ignored, which is why their proportion decreases over training.
> > > > >
> > > > > ### **Response to Q3: TAPO's convergence properties and stability**
> > > > >
> > > > > Please see response to W2 (4).

---

> > > > > > ### Author Response · Authors · 2025-11-28
> > > > > >
> > > > > > Thank you for your time and efforts in reviewing our work and for your insightful questions. Following your comments, we have provided detailed responses and revised the manuscript accordingly.
> > > > > >
> > > > > > Please let us know whether our responses address your concerns, and if any further clarification is needed.
> > > > > >
> > > > > > Best regards,
> > > > > >
> > > > > > Authors

---

### Official Review · Reviewer_jPh8 · 2025-11-01

**Soundness:** 3
**Presentation:** 3
**Contribution:** 3
**Rating:** 6
**Confidence:** 4

**Summary:**

This paper introduces a novel reinforcement learning framework named ReVeal, which aims to enhance the long-horizon reasoning and self-correction capabilities of Large Language Models (LLMs) in code generation tasks. The core contribution of this work lies in decomposing complex programming competition tasks into an iterative, multi-turn process of "code generation" and "test case generation." For this process, the authors designed a joint reward mechanism that includes Turn-Aware Policy Optimization (TAPO) to collaboratively optimize the model's code generation and self-verification abilities, thereby enabling deeper test-time expansion.

**Strengths:**

• Utilizing reinforcement learning to enhance a model's ability to solve complex problems, self-reflect, and correct code is a crucial research direction.

• The paper is well-structured with a clear and logical flow.

**Weaknesses:**

• The experimental dataset is somewhat limited: The main experiments are based on programming competition-style problems. While this effectively tests the model's algorithmic capabilities, it raises questions about the method's generalizability to broader, more realistic real-world development scenarios.

• Some experimental setups are not sufficiently clear or systematic: The experimental comparison section has issues that could affect the reliability of the conclusions, such as inconsistent evaluation turns and a lack of fairness in baseline comparisons. This makes it difficult for readers to accurately determine the source of the performance improvements.

**Questions:**

To further enhance the paper's persuasiveness and rigor, I suggest the authors consider the following points for revision and supplementation:
1. Recommendation to Expand Datasets to Verify Generalization
To more comprehensively validate the effectiveness and generalizability of the ReVeal framework, I strongly recommend that the authors supplement their experiments on more diverse datasets. For example: Foundational code generation datasets like HumanEval+ and MBPP+. Datasets that are closer to real-world development scenarios, such as CodeUJB or SWE-bench. Experiments on these datasets would demonstrate that ReVeal is not only applicable to algorithmic problems but can also generalize to complex, real-world software engineering scenarios, which would significantly enhance the value of this work.
2. Recommendation to Systematize Experimental Setup for Fair Comparison
Regarding the systematicity and fairness of the experimental setup, I have two main concerns:
Unify evaluation metrics for direct comparison: In the main results table (e.g., Table 1), the evaluation turns should be unified for all multi-turn methods being compared (e.g., reporting Pass@1 at turns 1, 5, and 8 for all methods). This would allow readers to more intuitively see the performance differences between methods at equivalent inference costs.
Configure a fair reasoning mechanism for key baselines: Baseline models like DAPO-Qwen2.5-32B and Single-turn RL should also employ a multi-turn feedback mechanism for test-time generation, similar to ReVeal. This would ensure a fair evaluation and help clearly distinguish whether the performance gains come from the RL training itself or from the reflection mechanism.
3. Report and Align Training Costs
The paper lacks a discussion of training costs. I recommend supplementing the paper with data on training overhead, such as the GPU hours for both ReVeal and Single-turn RL. This would not only help readers assess the method's cost-effectiveness but also allow for a fairer measurement of the ReVeal framework's true value and efficiency when compared under a similar training budget.

---

> ### Author Response · Authors · 2025-11-24
> **Response by authors [1/2]**
>
> We thank the reviewer for the constructive feedback and for recognizing the importance of our problem and the clarity of our paper. We address the concerns below.
>
> ### **Response to W1: Generalizability to more realistic scenarios**
> We agree that extending to more realistic real world development scenarios is a highly valuable application direction, and we will clarify this as a promising direction for future work. However, this is not the main focus of the current paper. Our main contribution is to propose a new RL training and inference paradigm that leverages reliable self verification to enable deeper test time scaling. We therefore use competitive programming problems as training data to study
> verification driven multi turn reasoning is reliable and scalable, and to validate the effectiveness of our method.
>
> At the same time, the ReVeal training paradigm, that is multi turn generation-verification framwork together with TAPO credit assignment, is a general concept and is not specific to competitive programming tasks. Whenever there exists a verifiable signal for candidate solutions, one can instantiate ReVeal by (i) defining an appropriate verification reward, (ii) wiring this reward into the TAPO return, and (iii) designing task specific tools and feedback.
>
> ### **Response to W2: Concern about systematic comparisons**
> This is addressed in our response to Question 2; briefly, we will unify evaluation turns and evaluate baselines under the same multi-turn framework for fair comparison.
>
> ### **Response to Q1: Generalizability to more datasets**
> We appreciate the suggestion to evaluate on more diverse datasets. In the revision (Appendix I), we have added experiments on HumanEval+ and MBPP+, using the same ReVeal-trained models and baselines. Although these benchmarks differ from our training data, ReVeal still yields larger gains over the baselines, suggesting that the benefits of ReVeal are not limited to competition-style problems.
>
> | Model              | HumanEval+ Pass@1 | MBPP+ Pass@1 |
> |--------------------|-------------------|--------------|
> | Qwen2.5-32B-Instruct | 85.4             | 75.4         |
> | DAPO-Qwen2.5-32B     | 86.0             | 73.8         |
> | Single-turn RL       | 86.0             | 76.2         |
> | ReVeal×8 (1 turn)    | 85.2             | 77.9         |
> | ReVeal×8 (3 turns)   | **88.6**         | **78.9**     |
> | ReVeal×8 (4 turns)   | 88.4             | **78.9**     |
>
>
> For benchmarks that are closer to real-world development, such as CodeUJB and SWE-bench, the supported programming languages and tool/execution environments differ from those used in our current setup, so we cannot directly evaluate our existing models. Running a full RL pipeline on these benchmarks would require substantial additional engineering (training data preparation, environment integration, execution sandboxing), which is beyond the scope of this submission. That said, we agree that extending ReVeal to such realistic software engineering benchmarks is an important and exciting direction for future work.

---

> > ### Author Response · Authors · 2025-11-24
> > **Response by authors [2/2]**
> >
> > ### **Response to Q2: Systematic comparisons**
> > We thank the reviewer for the constructive suggestion. we report Pass@1 at 1/5/8 turns for the ReVeal multi-turn experiments in the table below (see Appendix G); CTRL-related [1] results were not reported in the original paper, so we do not include them here.
> >
> > | Model  | LiveCodeBench V6 Pass@1  | CodeContests Pass@1  |
> > |-------------------------------|----------------------------------------------|--------------------------------------------|
> > | ReVeal (outcome only)  | 32.7@turn1 → 35.9@turn5 → 36.1@turn8  | 20.0@turn1 → 26.1@turn5 → 27.4@turn8  |
> > | ReVeal TAPO with joint rewards | 34.8@turn1 → 37.2@turn5 → 37.7@turn8  | 22.3@turn1 → 28.1@turn5 → **30.4**@turn8  |
> >
> >
> > We thank the reviewer for pointing out the need to disentangle the contribution of multi-turn framwork from RL training. In the original submission (Appendix C, Table 6), we included a ReAct-style multi-turn prompting baseline (i.e. ReVeal framwork with same prompt and tool w/o training) on Qwen2.5-32B-instruct, aimed at disentangling the benefit of the multi-turn framework from that of RL training. In the revision, we refine the description and consolidate it with new ablations in Appendix F. In addition, we apply the same multi-turn generation-verification framework with code execution tools to key baselines such as DAPO-Qwen2.5-32B and Single-turn RL at test time, and add these results in Appendix F. These multi-turn variants either exhibit gains in the first few turns and then begin to degrade, or show no improvement at all, indicating that while the multi-turn framework and tool feedback are helpful, the main gains come from explicitly training with the ReVeal RL objective.
> >
> > | Configuration                                              | LiveCodeBench V6 Pass@1 ($\Delta_{\uparrow}$, $\Delta_{\downarrow}$)                                          |
> > |-----------------------------------------------------------|----------------------------------------------------------------------------------------------------------------|
> > | DAPO-Qwen2.5-32B + ReVeal multi-turn framework                  | 28.4@turn1 → 32.8@turn5 ($\Delta_{\uparrow}$=4.53, $\Delta_{\downarrow}$=0.75) → 31.8@turn8 ($\Delta_{\uparrow}$=4.15, $\Delta_{\downarrow}$=1.70) |
> > | Single-turn RL model + ReVeal multi-turn framework        | 33.4@turn1 → 32.8@turn5 ($\Delta_{\uparrow}$=1.53, $\Delta_{\downarrow}$=1.36) → 33.0@turn8 ($\Delta_{\uparrow}$=2.05, $\Delta_{\downarrow}$=1.53) |
> > | ReVeal (full RL)                                          | 34.8@turn1 → 37.2@turn5 ($\Delta_{\uparrow}$=3.71, $\Delta_{\downarrow}$=0.0) → 37.7@turn8 ($\Delta_{\uparrow}$=5.62, $\Delta_{\downarrow}$=0.0)   |
> >
> > **Reference**
> >
> > [1] Teaching Language Models to Critique via Reinforcement Learning
> >
> > ### **Response to Q3: Discussion of training Costs**
> > We thank the reviewer for pointing out the missing discussion of training costs. We report training time measured on a single B200 node with 8 GPUs. Under this setting, Single-turn RL takes about 48.3 hours, while ReVeal RL takes about 112.3 hours. The additional cost mainly comes from inferencing multi-turn (3-turn) trajectories during training. These numbers are measured using the VERL implementation, which executes the three turns sequentially within each batch and does not yet use asynchronous multi-turn rollouts. With asynchronous rollout optimizations, where the next-turn generation for a sample can start as soon as its previous turn finishes, the overhead of ReVeal can be further reduced.

---

> > > ### Author Response · Authors · 2025-11-28
> > >
> > > Thank you for your time and efforts in reviewing our work and for your insightful questions. Following your comments, we have provided detailed responses and revised the manuscript accordingly.
> > >
> > > Please let us know whether our responses address your concerns, and if any further clarification is needed.
> > >
> > > Best regards,
> > >
> > > Authors

---

### Official Review · Reviewer_utHn · 2025-11-01

**Soundness:** 2
**Presentation:** 3
**Contribution:** 3
**Rating:** 6
**Confidence:** 3

**Summary:**

Current Reinforcement Learning with Verifiable Rewards (RLVR) only optimizes "generation capability" relatively well, while "verification capability" is barely explicitly optimized—leading to a significant gap between generation and verification—this paper proposes ReVeal, a multi-turn reinforcement learning (RL) framework for code agents. Within a single trajectory, ReVeal alternates between "generation → verification → tool feedback" and, for the first time, places "self-verification" and "generation" under the same optimization target. In this process, the model not only generates code but also automatically produces executable test cases and leverages feedback from external tools for self-correction. To ensure the balanced improvement of generation and verification capabilities, ReVeal introduces Turn-Aware Policy Optimization (TAPO), which assigns explicit reward signals to both generation and verification actions at each turn. This enables fine-grained credit assignment, thereby simultaneously enhancing these two capabilities during the training phase and narrowing the performance gap between generation and verification. Unlike traditional sparse reward methods that rely solely on the final pass rate, ReVeal incorporates tool-based feedback from interpreters/code judges at each turn. This allows the model to continuously optimize code quality during the inference phase, even when performing a large number of inference turns. Experimental results demonstrate that ReVeal achieves excellent performance on LiveCodeBench, validating the effectiveness of optimizing verification capability on par with generation capability.

**Strengths:**

1.	The ReVeal framework innovatively introduces explicit verification turns during the training process and uses the gold solution to pre-verify the generated test cases. This method enables the model to explicitly learn self-verification capabilities during training, thereby enhancing the reliability of self-verification.
2.	ReVeal introduces a Turn-level return mechanism: once the generation is correct, the test generator of the previous round can also receive rewards. This design effectively ties the generation and verification processes together, preventing undesirable behaviors such as "writing useless tests" and other reward gaming phenomena, thus promoting the co-optimization of the generation and verification processes.
3.	Experimental results show that on LiveCodeBench V6, as the number of inference turns increases, the model's code generation capability and performance continue to improve. This verifies the phenomenon that even with short training turns, the model can still perform continuous multi-turn reasoning during inference. The model indeed learns effective self-verification strategies during training, enabling it to continuously improve its performance in the inference phase.

**Weaknesses:**

1.	The paper motivates ReVeal as a way to make self-verification reliable “in realistic environments where public tests are unavailable”. During training, however, the model-generated tests are filtered against a golden solution to guarantee high-quality feedback. At evaluation time (e.g., LiveCodeBench) the final correctness is judged by the benchmark’s own test suites, i.e., still under a setting with reliable canonical tests. Thus, the experiments mainly demonstrate that optimizing verification helps on code benchmarks that already provide trustworthy execution oracles, rather than showing robustness in the fully autonomous, low-test-coverage setting highlighted in the introduction. This is not fatal — code is exactly the domain where such oracles exist — but the paper’s claim about “realistic environments without pre-existing tests” would be stronger with a no-filter / noisy-test ablation.
2.	Current experiments are only conducted on large 32B models, and do not involve the performance of small-scale models such as 7B or 14B, nor sparse models. Considering that small-scale models may face greater challenges in generating test cases, the lack of verification on small-scale/sparse models limits the generalization ability of the paper's conclusions regarding model scale. It is suggested that the authors expand the experimental scope in future research to cover models of different scales, so as to evaluate the applicability of the ReVeal framework on various models.
3.	The paper provides a relatively brief description of the reward mechanism, and does not elaborate on the specific calculation methods and weight settings of format rewards, final pass rewards, turn-level rewards, etc. This may require researchers to adjust these hyperparameters by themselves when reproducing the experiments. It is suggested that the authors provide more detailed reward mechanism design and hyperparameter settings or released the open-source implementation of the ReVeal framework in future research to improve the transparency and reproducibility of the study.

**Questions:**

1.	The turn-level return (Eq. 6 in the manuscript) assigns each generation reward to both the generation turn and the preceding verification turn with weight 1.0. This tightly couples the verification reward to the next-turn generation quality, which can increase variance when the next turn underperforms (e.g., due to context truncation or large code edits). Have the authors tried attenuated backflow or exponential decay across later turns to test whether verification still resists reward hacking under weaker coupling? Even a small ablation would clarify whether the current choice of weight 1 is essential or merely convenient.
2.	Since the policy is shared between generation and verification, it is hard to disentangle whether the gain comes from (i) better code generation under multi-turn tool feedback, or (ii) explicit optimization of the verification subtask. A simple ablation where the verification head (or the verification part of the loss) is frozen/removed, while keeping the rest of the setup identical, would clarify the contribution of “training verification as a first-class task”. Without such an ablation, the current evidence could still be explained by “multi-turn generation with tool context”.
3.	A key claim is that ReVeal, by explicitly optimizing verification, can extrapolate beyond the training horizon (3 turns → 25 turns at inference). This is compelling, but currently only one training horizon is reported. Could the authors report results with a longer training horizon (e.g., 5 turns) and a shorter one (e.g., 1–2 turns) to show (i) whether the benefit saturates, and (ii) whether the ability to extrapolate is truly due to the proposed TAPO design rather than a coincidental choice of 3 turns? This is especially relevant since context is truncated to the last few turns during inference.

---

> ### Author Response · Authors · 2025-11-24
> **Response by authors [1/3]**
>
> We thank the reviewer for the constructive feedback and for recognizing that ReVeal offers a novel perspective by explicitly optimizing verification as a first-class objective alongside generation, with a principled turn-level credit assignment mechanism. We address the concerns point by point below.
>
> ### **Response to W1: Clarification on “realistic environments without public tests” and oracle usage**
>
> We agree that our experiments are conducted on code data with oracles, since such oracles are necessary for RLVR training and evaluation. However, our setup is specifically designed to **mimic the realistic scenario where users do not provide ready-made tests to the model at inference time**.  By “public tests” we mean test cases that are exposed to the model at inference time (e.g., the public tests in CodeContests provided to the model in RLEF [1] and other related work). In contrast, ReVeal never sees such public tests at inference: on both LiveCodeBench and CodeContests, the model only sees the problem description and will fully autonomously generate and execute its own tests **without any filtering**.
>
> During training we follow the **RLVR paradigm**, using golden code and tests to compute accurate verifiable rewards. The filtering of test cases by golden code affects only the <tool-feedback> , we avoid executing wrong tests on candidate code to keep the feedback accurate and better guide subsequent solution exploration during training, but **all generated tests remain in the trajectory and participate in training.** In this sense, we **use the oracle only to train a stronger verifier and obtain reliable feedback** at inference time without depending on pre-existing public tests as some baselines do. RL with noisy or incorrect test feedback is an interesting direction, but it is beyond the scope of this work, as our goal is precisely to learn a _reliable_ verification that can provide accurate feedback and enable deeper refinement at inference.
>
> **Reference**
>
> [1] RLEF: Grounding Code LLMs in Execution Feedback with Reinforcement Learning
>
> ### **Response to W2: generalization to small-scale models**
>
> We appreciate the reviewer’s suggestion. Following this feedback, we additionally evaluated ReVeal on a smaller model, Qwen3-4B-Instruct. The results show that ReVeal brings substantial performance gains, far exceeding other baseline methods. This provides further evidence that ReVeal remains highly effective even for small models.
>
> | Model                    | LiveCodeBench V6 Pass@1                  | CodeContests Pass@1                     |
> |--------------------------|-------------------------------------------|-----------------------------------------|
> | Qwen3-4B-Instruct        | 33.1                                     | 24.3                                    |
> | Single-turn RL           | 39.0                                     | 26.9
> | ReVeal| 40.6@turn1 → 44.1@turn5 → **44.5**@turn8 | 28.5@turn1 → 33.6@turn5 → **33.9**@turn8 |
>
>
> The corresponding results are now included in Appendix E.
>
> ### **Response to W3: Clarification on Reward Calculation and Hyperparameter Settings**
>
> In Sec. 2.2.1 of the revision, we added an explanation of the format reward: it (i) enforces the prescribed generation-turn / verification-turn tags so that we can correctly identify each turn and assign turn-level rewards, and (ii) constrains the code and test-case format, especially the test input/output blocks, so that we can reliably extract code and tests for tool execution.
>
> In Sec. 2.2.1, we already provided the weights and formulas (Eq. 1–3) for all the rewards: the outcome pass-rate reward (weight 5), the generation reward derived from the same pass-rate signal (thus scaled by 5), the verification reward (weight 1), and the format reward (weight 1). These weights are fixed across all experiments.
>
> To further improve transparency and reproducibility, we have shared the code in the last reply.

---

> > ### Author Response · Authors · 2025-11-24
> > **Response by authors [2/3]**
> >
> > ### **Response to Q1: Coupling between verification and next-turn generation**
> > First we'd like to clarify that the weight 1.0 in Eq. (6) only controls how strongly the generation reward of turn k+1 is credited back to the preceding verification turn k; it does **not** correspond to leaking verification reward into generation tokens, which is what we refer to as reward hacking. In that case, code tokens would be updated by a signal they do not causally influence (for example, wrong code but strong tests still giving high verification reward to the code, or correct code but weak tests inducing low verification reward), and the policy may learn spurious correlations between code patterns and high test accuracy. TAPO explicitly avoids this hacking by never assigning $r_{ver}$ to generation tokens.
> >
> > The suggested issues (e.g., context truncation or large code edits causing underperformance of the next turn and incorrectly propagating $r_{gen}$​ back to the preceding verification turn) are rare in our actual runs: empirically, once verification accuracy becomes high, the rate of correction vs. degradations ($\Delta_{\uparrow}$  vs. $\Delta_{\downarrow}$ in Table 1) shows that ReVeal learns to use tool feedback for **targeted corrections**, with a high rate of successful fixes and a very low rate of harmful edits. In contrast, baselines without verification training have much higher degradation rates, which indicates that **the next-turn code quality is usually aligned with the preceding verifciation accuracy** rather than dominated by randomness. Context truncation is solved by clipping overlong responses. Moreover, in the ReVeal outcome-only baseline, verification tokens receive the outcome reward (this coupling weight is also 1), and **this coupling exactly drives the improvement in test-case accuracy** (Fig. 4 (d)). TAPO only adds a **more local and lower-variance dependency by routing the next-turn generation reward back only to the immediately preceding verification turn**. We have not run a systematic sweep over this weight, mainly due to RL cost and the fact that the current setting already gives stable training and clear gains.
> >
> > ### **Response to Q2: Separating framework-level and RL training for verification gains**
> > We thank the reviewer for pointing out the need to disentangle the contribution of multi-turn framwork with tool from explicit optimization of verification. In the original submission (Appendix C, Table 6), we included a ReAct-style multi-turn prompting baseline (i.e. ReVeal framwork with same prompt and tool) on Qwen2.5-32B-instruct, aimed at disentangling the benefit of the multi-turn framework from that of RL training. In the revision, we refine the description and consolidate it with new ablations in Appendix F.
> >
> > We interpret the suggestion of “removing the verification part of the loss” as masking the RL loss for all verification tokens. This would remove all learning signal for verification, including the format, and would also hurt the optimization of later generation turns because they would receive poor feedback from the verification. Instead, we provide a lower-cost and stronger variant, **ReVeal (no-verification RL)**, where we apply the ReVeal multi-turn framework and tools, use the **ReVeal-trained model as the generator and untrained base model as verifier**. The results show that this variant yields gains in the first 5 turns and quickly saturates, whereas the full ReVeal model continues to improve with more turns with higher $\Delta_{\uparrow}$ and lower $\Delta_{\downarrow}$, indicating that ReVeal’s explicit optimization of verification yields clear performance gains and enables deeper test-time scaling.
> >
> > | Configuration                                              | LiveCodeBench V6 Pass@1 ($\Delta_{\uparrow}$, $\Delta_{\downarrow}$)                                          |
> > |-----------------------------------------------------------|----------------------------------------------------------------------------------------------------------------|
> > | ReVeal (no-ver RL): ReVeal as Generator + Base model as Verifier | 34.6@turn1 → 36.9@turn5 ($\Delta_{\uparrow}$=2.94, $\Delta_{\downarrow}$=0.53) → 36.8@turn8 ($\Delta_{\uparrow}$=4.25, $\Delta_{\downarrow}$=1.26) |
> > | ReVeal (full RL)                                          | 34.8@turn1 → 37.2@turn5 ($\Delta_{\uparrow}$=3.71, $\Delta_{\downarrow}$=0.0) → 37.7@turn8 ($\Delta_{\uparrow}$=5.62, $\Delta_{\downarrow}$=0.0)   |

---

> > > ### Author Response · Authors · 2025-11-24
> > > **Response by authors [3/3]**
> > >
> > > ### **Response to Q3: Training horizon and the source of extrapolation ability**
> > > We thank the reviewer for this suggestion. In Appendix H, we vary the training horizon to 1, 3, and 5 turns while keeping other settings fixed. All ReVeal variants outperform the single-turn RL baseline; training with 3 turns yields gain over 1-turn training, while increasing the horizon further to 5 turns brings no additional improvement, suggesting that the benefit saturates around 3-5 turns on our current (mostly solvable) training set. On this training set, mean rewards can reach 4.5, which means most problems are already solved within 3 generation-verification turns, so the additional 2 training turns provide limited extra signal on a small fraction of very difficult questions. We expect longer training horizons to be more beneficial on more challenging training data.
> > >
> > > | Model                    | LiveCodeBench V6 Pass@1                  | CodeContests Pass@1                     |
> > > |--------------------------|-------------------------------------------|-----------------------------------------|
> > > | Qwen3-4B-Instruct        | 33.1                                     | 24.3                                    |
> > > | Single-turn RL           | 39.0                                     | 26.9                                    |
> > > | ReVeal (max train turn=1)| 37.3@turn1 → 41.5@turn5 → 41.7@turn8     | 26.6@turn1 → 30.6@turn5 → 30.7@turn8   |
> > > | ReVeal (max train turn=3)| 40.6@turn1 → 44.1@turn5 → **44.5**@turn8 | 28.5@turn1 → 33.6@turn5 → **33.9**@turn8 |
> > > | ReVeal (max train turn=5)| 38.6@turn1 → 43.1@turn5 → 44.0@turn8     | 28.4@turn1 → 33.2@turn5 → 33.5@turn8   |
> > >
> > >
> > > Importantly, for all training horizons (1, 3, and 5), the corresponding models continue to improve when we allow more inference turns than they were trained on, indicating that extrapolation beyond the training horizon is not tied to the specific choice of 3 turns. Moreover, Appendix C shows that truncating the context to the last 3 turns vs. using the full history yields similar Pass@1 at turn 15 (38.2% vs. 38.3%), showing that the number of turns visible at inference time has little impact on the results. These results support that the extrapolation behavior stems from the ReVeal framework and TAPO design.

---

> ### Author Response · Authors · 2025-11-24
> **Response by authors of ReVeal Code Link**
>
> We have shared the code in this link, https://anonymous.4open.science/r/iclr_anonymous-7838.

---

> > ### Author Response · Authors · 2025-11-28
> >
> > Thank you for your time and efforts in reviewing our work and for your insightful questions. Following your comments, we have provided detailed responses and revised the manuscript accordingly.
> >
> > Please let us know whether our responses address your concerns, and if any further clarification is needed.
> >
> > Best regards,
> >
> > Authors

---

### Author Response · Authors · 2025-11-24
**General Response**

Dear Reviewers,

We sincerely appreciate your time in reviewing our paper and for the many constructive comments. We are encouraged that several reviewers found ReVeal to present a **genuinely novel perspective by explicitly optimizing verification as a first-class objective with a principled turn-level credit assignment mechanism** (utHn, beFR), and that you highlighted the **importance and originality** of the problem and our approach (jPh8, 6pm9, beFR), as well as the **thoroughness of the empirical evaluation** (6pm9). In the revision, we have focused on addressing the main shared concerns:

**Separating framework-level gains from RL training.** Under the same multi-turn generation–verification framework, we add several variants where RL is disabled for either generation or verification or both. Variants with RL disabled for verification show only limited and quickly saturating gains, while full ReVeal and variants with RL disabled for generation continue to improve with more turns. _(for utHn, jPh8, 6pm9; Appendix F)_

**Ablations on TAPO’s reward design.** We add ReVeal outcome+gen and ReVeal outcome+ver variants, together with comparisons of different absolute/improvement coefficients in the generation reward. _(for beFR, 6pm9; Appendix G)_

**Generalization across model scales.** We add experiments on the smaller Qwen3-4B-Instruct model, where ReVeal still brings substantial gains over strong baselines. _(for utHn, beFR; Appendix E)_

**Evaluation on additional benchmarks.** Using the same trained models, we evaluate on HumanEval+ and MBPP+, where ReVeal again outperforms baselines, indicating that ReVeal generalizes beyond competition-style datasets. _(for jPh8, beFR; Appendix I)_

**Training-horizon ablation.** We compare models trained with 1/3/5 turns and show that all of them extrapolate beyond their training horizon at inference time. _(for utHn; Appendix H)_

**Gradient-based incentive analysis of TAPO.** We provide a gradient-level comparison between TAPO and outcome-only PPO, clarifying how TAPO strengthens optimization pressure on verification while preventing reward hacking, without changing the underlying PPO optimizer. _(for beFR, 6pm9, utHn; Appendix J)_

All of these additions are incorporated into the revised manuscript and highlighted for your convenience. We hope they address the shared concerns and help clarify the scope and contributions of our work.

Sincerely,
Authors

---

### Comment · Area_Chair_ha8G · 2025-11-25

Dear Reviewer, thank you for reviewing for ICLR. Since the discussion deadline is coming soon, could you please take a look at the author's rebuttal, respond to their comments, and update your rating as well? Thanks!

---

### Meta-Review · Area_Chair_JUYF · 2026-01-06

**Summary:**

1. The degree of realism that the oracle-based test setup guarantees.
2. Breath of the experimental comparison.
3. No theory, paper purely empirical.
4. The framework does not consider the memory/time complexity of the solution identified by the code agent.

**Reviewer Concerns:**

1. The degree of realism was largely addressed.
2. Additional experiments were provided.
3. No theory is justified in a purely applied paper (better do it this way than add "mathy" theory).
4. A correctness-only framework is justified given the state of the field now.

**Reviewer Scores:**

1. 6pm9: 2 -> 4 or 6 (the reviewer explicitly said concerns were addressed)
2. beFR: 4-> 6 (ablations were provided)
3. Other reviewers: no change (none seemed so enthusiastic as to give it an 8)

---

### Decision · Program_Chairs · 2026-01-26

Accept (Poster)